



# Contribution of combustion Fe in marine aerosols over the northwestern Pacific estimated by Fe stable isotope ratios

Minako Kurisu[1], Kohei Sakata[2], Mitsuo Uematsu[3,4], Akinori Ito[5], and Yoshio Takahashi[6]

[1]Research Institute for Global Change, Japan Agency for Marine-Earth Science and Technology (JAMSTEC),
2-15, Natsushima-cho, Yokosuka, Kanagawa, 237-0061, Japan
[2]Center for Global Environmental Research, National Institute for Environmental Studies, 16-2 Onogawa, Tsukuba, Ibaraki
305-8506, Japan
[3]Atmosphere and Ocean Research Institute, The University of Tokyo, Chiba 277-8564, Japan
[4]Center for Environmental Science in Saitama, Saitama 347-0115, Japan
[5]Research Institute for Global Change, JAMSTEC, Yokohama, Kanagawa, 236-0001, Japan
[6]Department of Earth and Planetary Science, Graduate School of Science, The University of Tokyo, 7-3-1 Hongo, Bunkyo-ku,
Tokyo 113-0033, Japan

*Correspondence to*: Minako Kurisu (kurisum@jamstec.go.jp)

**Abstract.** The source apportionment of aerosol iron (Fe), including natural and combustion Fe, is an important issue because aerosol Fe can enhance oceanic primary production in the surface ocean. Based on our previous finding that combustion Fe emitted by evaporation processes has Fe isotope ratios ($\delta^{56}$Fe) that are approximately 4 ‰ lower than those of natural Fe, this study aimed to distinguish aerosol Fe sources over the northwestern Pacific using two size-fractionated marine aerosols. The $\delta^{56}$Fe values of fine and coarse particles from the eastern or northern Pacific were found to be similar to each other, ranging from 0.0 to 0.4 ‰. Most of them were close to the crustal average, suggesting the dominance of natural Fe. On the other hand, particles from East Asia demonstrated lower $\delta^{56}$Fe values in fine particles ($-0.5$ to $-2.2$ ‰) than in coarse particles (on average 0.1 ‰). The correlations between the $\delta^{56}$Fe values and the enrichment factors of lead and vanadium suggested that the low $\delta^{56}$Fe values obtained were due to the presence of combustion Fe. The $\delta^{56}$Fe values of the soluble component of fine particles in this region were lower than the total, indicating the preferential dissolution of combustion Fe. In addition, we found a negative correlation between the $\delta^{56}$Fe value and the fractional Fe solubility in air masses from East Asia. These results suggested that the presence of combustion Fe is an important factor in controlling the fractional Fe solubility in air masses from East Asia, whereas other factors were more important in the other areas. By assuming typical $\delta^{56}$Fe values for combustion and natural Fe, the contribution of combustion Fe to the total (acid-digested) Fe in aerosols was estimated to reach up to 50 % of fine and 21 % of bulk (coarse + fine) particles in air masses from East Asia, whereas its contribution was small in the other areas. The contribution of combustion Fe to the soluble Fe component estimated for one sample was approximately twice as large as the total, indicating the importance of combustion Fe as a soluble Fe source, despite lower emissions than the natural. These isotope-based estimates were compared with those estimated using an atmospheric chemical transport model (IMPACT), in which the fractions of combustion Fe in fine particles, especially in air masses from East Asia, were consistent with each other. In contrast, the model estimated a relatively large contribution from combustion Fe in coarse particles, probably because of the different characteristics of combustion Fe that are included in the model calculation and the isotope-based estimation. This highlights the importance of observational data of $\delta^{56}$Fe for size-fractionated aerosols to scale the combustion Fe emission by the model. The average deposition fluxes of soluble Fe to the surface ocean were 1.4 and 2.9 nmol m$^{-2}$ day$^{-1}$ from combustion and natural aerosols, respectively, in air masses from East Asia, which suggests combustion Fe could be an important Fe source to the surface seawater among other Fe sources. Distinguishing Fe sources using the $\delta^{56}$Fe values of marine aerosols and seawater is anticipated to lead to a more quantitative understanding of the Fe cycle in the atmosphere and surface ocean.



## 1. Introduction

45        Iron (Fe) is an essential element for marine biota. Although Fe is the fourth most abundant element in the Earth's crust (Taylor, 1964), the concentration of Fe in the surface ocean can be as low as 0.05 nM (Martin et al., 1989; Schlitzer et al., 2018). In high-nutrient low-chlorophyll (HNLC) regions, the deficiency of bioavailable Fe is considered one of the main limiting factors of primary production of phytoplankton (Martin et al., 1994; Martin and Fitzwater, 1988; Moore et al., 2013). Phytoplankton activity has a major impact on the biogeochemical cycles of various elements, such as carbon, nitrogen, sulfur,

and other trace elements (Charlson et al., 1987; Falkowski et al., 1998). In particular, ocean Fe fertilization plays an important role in the uptake of carbon dioxide ($CO_2$), decreasing the partial pressure of $CO_2$ in the atmosphere and positively affecting the global climate (Ciais et al., 2013; Falkowski et al., 1998). Therefore, understanding how Fe is supplied to the surface ocean is an important issue.

       Atmospheric aerosols, mainly in the form of mineral dust, are considered to be one of the main components that

supply Fe to the surface ocean. Mineral dust transported from East Asia has been recognized as an important supplier of Fe to the surface of the North Pacific ocean, including HNLC regions, although there is significant seasonal variation in the flux of mineral dust (Duce et al., 1991; Jickells, 2005; Uematsu et al., 1983). In addition to aerosols, the dissolution of Fe from coastal sediments (Lam and Bishop, 2008; Nishioka et al., 2013; Nishioka and Obata, 2017) and hydrothermal vents (Fitzsimmons et al., 2017; Tagliabue et al., 2010) have been suggested to be important sources of dissolved Fe in the surface ocean.

60        A great deal of attention has been paid to combustion Fe in aerosols as an important source of Fe in the surface ocean (Ito et al., 2019; Kurisu et al., 2016b, 2019; Sholkovitz et al., 2009). The amount of Fe emitted globally from combustion sources is estimated to be 2.1±0.5 TgFe yr$^{-1}$, which is an order of magnitude smaller than the mineral dust emission (72±43 TgFe yr$^{-1}$; Myriokefalitakis et al., 2018). However, combustion Fe is considered to be an important source of soluble Fe, which is thought to be easily bioavailable, because of its high Fe fractional solubility in comparison with that of Fe in mineral dust

(Schroth et al., 2009; Sholkovitz et al., 2009; Takahashi et al., 2013). Moreover, the fertilizing effect of combustion Fe has been increasingly examined in ocean biogeochemistry models, some of which showed combustion Fe is more efficient at enhancing marine productivity than Fe from mineral dust sources (Hamilton et al., 2020; Ito et al., 2020b). It is, however, still a major challenge to quantitatively understand and predict the high variability and complex aerosol chemistry of combustion Fe sources and the diverse marine biogeochemical responses (Ito et al., 2021).

70        The fractional Fe solubility is defined as follows:

$$fractional\ Fe\ solubility\ (\%) = \frac{S-Fe}{T-Fe} \times 100, \tag{1}$$

where T-Fe and S-Fe are the concentrations of acid-digested (total) and soluble Fe, respectively (Buck et al., 2006; Morton et al., 2013). The fractional Fe solubility of aerosols is controlled by various factors, such as (i) the size or surface area of the particles (Baker and Jickells, 2006), (ii) the occurrence of atmospheric reactions with various acids (of both natural and

anthropogenic origin) and photochemical reactions during transportation (Chen and Grassian, 2013; Ito et al., 2019; Takahashi et al., 2011), and (iii) differences in the chemical forms of Fe. Variations in the chemical form of Fe depend heavily on the source of Fe. Iron in mineral dust is mainly found in the form of crystalline Fe oxides or Fe in aluminosilicates, which typically exhibit very low fractional solubility (approximately 0.1 to 1 %). Combustion aerosols contain more labile forms of Fe, such as Fe (hydr)oxides, aggregates of nano-sized Fe oxides, and Fe sulfates. They are highly soluble (up to 80 %) or more easily

solubilized by atmospheric reactions (Schroth et al., 2009; Takahashi et al., 2011). However, the relative importance of these factors remains unclear, particularly in the marine atmosphere.

       The contribution of combustion and natural Fe in aerosols has been estimated in various modeling studies (e.g., Ito et al., 2019; Luo et al., 2008; Matsui et al., 2018; Myriokefalitakis et al., 2018), the results of which suggest the important contribution of combustion Fe to soluble Fe deposition (up to 90 % depending on areas and models). The parameterizations

used in these studies differ depending on the model used, especially in terms of the amount of Fe emitted from various sources,





the Fe solubility at emission (particularly for combustion Fe), and the solubilization processes that occur during atmospheric transportation, which lead to different results. Myriokefalitakis et al. (2018) compared the results of several model calculations with observational data and reported several problems, including (i) different estimations of the contribution of combustion Fe to remote ocean regions among the various models, (ii) an overestimation of the concentration of atmospheric Fe near dust

source regions, and (iii) an underestimation of the Fe concentrations in remote oceanic regions compared with observation. They claimed that some uncertainties remain, such as the relative fraction of combustion and dust Fe to the soluble Fe that is present over remote oceanic regions, aerosol size distributions, and atmospheric processing under the presence of combustion Fe and anthropogenic pollutants. These model studies usually evaluate their results by comparing the estimated Fe concentrations/solubilities with observed T-Fe and S-Fe concentrations. However, this method cannot be used to directly

compare the fractions from different sources of Fe.

The iron stable isotope ratio can be used to distinguish Fe from different sources (Dauphas et al., 2017). It is reported as the value of $\delta^{56}$Fe relative to a standard material (IRMM-014, Institute for Reference Material and Measurements), using the following equation:

$$\delta^{56}Fe(‰) = \left( \frac{(^{56}Fe/^{54}Fe)_{sample}}{(^{56}Fe/^{54}Fe)_{IRMM-014}} - 1 \right) \times 1000. \tag{2}$$

The average $\delta^{56}$Fe value of mineral dust is $-0.01\pm0.08$ ‰, which is similar to that of terrestrial igneous rocks or soil ($0.00\pm0.05$ ‰; Beard et al., 2003). In contrast, combustion Fe has a comparatively low $\delta^{56}$Fe value (as low as $-4$ ‰), which is a result of the kinetic isotope fractionation that occurs during evaporation under high-temperature conditions ($> 800$ ℃), such as fossil fuel combustion, industrial production of metals, and vehicle emissions, among others (Kurisu et al., 2016a, 2016b, 2019; Kurisu and Takahashi, 2019). Combustion Fe can be distinguished from natural Fe, which is not directly

associated with the observed T-Fe and S-Fe concentrations (Meskhidze et al., 2019), by using the isotope ratio as a tracer. Ascertaining the source of Fe in seawater using $\delta^{56}$Fe values has been conducted in several studies, with aerosols considered to be an important source of Fe, even in seawater (Conway and John, 2014; Labatut et al., 2014; Pinedo-González et al., 2020). Pinedo-González et al. (2020) observed low $\delta^{56}$Fe signals in the surface seawater of the North Pacific that may have originated from combustion Fe in East Asia. However, the $\delta^{56}$Fe of the aerosols was not measured during the same period in this study,

and there have been few reports concerning the collection of Fe isotope ratios from aerosols over the ocean. Conway et al. (2019) reported the Fe isotope ratios of marine aerosols over the North Atlantic and suggested that the total (acid-digested) aerosols contained a $\delta^{56}$Fe value close to the crustal value, whereas soluble aerosols had low $\delta^{56}$Fe values due to the presence of combustion Fe with low $\delta^{56}$Fe and high fractional Fe solubility. Our previous study reported a limited dataset of $\delta^{56}$Fe values of two-size-fractionated marine aerosols collected near the Japanese coast, which indicated that acid-digested fine particles had

had considerably lower values of $\delta^{56}$Fe than coarse particles because of the presence of combustion Fe in fine particles (Kurisu et al., 2016b). However, more data relating to the $\delta^{56}$Fe of aerosols collected in the northwestern Pacific, including the open ocean, are required to gain further insights into the atmospheric mixing among air masses of combustion Fe of various origins. Furthermore, size-fractionated aerosol sampling would make it possible to discuss the influence of combustion Fe more clearly because of the difference in the size distribution of combustion and natural aerosols.

In this study, two-size-fractionated aerosol samples were collected in the northwestern Pacific, to where aerosols from East Asia can be transported and the influence of combustion Fe can be observed. In this study, we aimed (i) to discuss the contribution of combustion and natural Fe to marine aerosols based on Fe isotope ratios and (ii) to evaluate the importance of combustion Fe as a controlling factor of fractional Fe solubility by combining data relating to the fractional solubility, Fe species, and Fe isotope ratios. We also compared the fraction of combustion Fe in marine aerosols estimated using Fe isotope

data with that estimated by a model calculation, with the aim of discussing the applicability of Fe isotopes to improving process-based estimations.



## 2. Methods

### 2.1. Sample collection

Marine aerosol samples were collected during the R/V *Hakuho Maru* KH-13-7 (12 December 2013 to 11 February
2014, samples 13-a to 13-e, Table S1) and KH-14-3 cruises (24 June 2014 to 9 August 2014, samples 14-A to 14-O, Table S2)
around the northwestern Pacific (Fig. 1). The aerosol sampling was conducted on the compass deck (13 m above sea level).
Two size-fractionated aerosols (finer and coarser than 2.5 μm) were collected on PTFE filters (ADVANTEC, PF040, 90 mm
φ) using a high-volume virtual dichotomous air sampler (Kimoto, Model AS-9, Japan) at a mean flow rate of 15 $m^3 h^{-1}$. The
sampler was automatically controlled by a wind sector that was operated only when the relative wind direction ranged from
−90° to 90° from the perspective of the bow and the relative wind speed was more than 1 m $s^{-1}$. The typical sampling duration
was three days and the mean volume of filtered air was approximately 720 $m^3$. The samples were stored at −18 °C until
chemical analysis could be conducted. Filter blanks were checked by setting filters on the sampler without running.

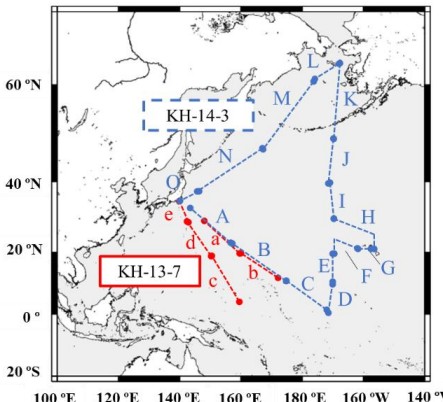

**Figure 1. Sampling areas in this study. The base figure was created using General Mapping Tools (GMT, Wessel et al., 2019).**

### 2.2. Acid digestion of aerosol samples

Acid digestion of the aerosol samples was conducted to determine the total concentrations of Fe and other trace
elements, including titanium (Ti), vanadium (V), and lead (Pb). All the procedures were conducted in a HEPA-filtered (SS-
MAC15, Air Tech, Japan) clean bench in a class-10000 clean room. Perfluoroalkoxy (PFA) vials (Savillex, USA) and low-
density polyethylene (LDPE) bottles (Nalgene, Thermo Fisher Scientific, Germany) were used for sample processing and
storage of the solutions, respectively. All equipment was preliminarily washed by sequential soaking overnight in warm 3 mol
$L^{-1}$ nitric acid ($HNO_3$, for the electronics industry, FUJIFILM Wako Pure Chemical Corp., Japan), warm 3 mol $L^{-1}$
hydrochloric acid (HCl, for the electronics industry, FUJIFILM Wako Pure Chemical Corp., Japan), and ultrapure water (18.2
MΩ<, Milli-Q, Millipore GmbH, Japan). All acids used for the digestion, including HNO3, HCl and hydrofluoric acid (HF)
were Tamapure AA-100 grade (each metallic impurity less than 100 pg $ml^{-1}$, Tama Chemical Co., LTD., Japan).
Approximately 1/16 of each sample filter was digested in a closed PFA vial by adding 2 mL of 15.3 mol $L^{-1}$ $HNO_3$, 2 mL of
9.3 mol $L^{-1}$ HCl, and 1 mL of 22 mol $L^{-1}$ HF. The sample in the vial was then heated at 150 ℃ for one day and evaporated to
near dryness. A 1 mL volume of 15.3 mol $L^{-1}$ $HNO_3$ was then added and the sample was again heated at 150 ℃ for one day
and evaporated to near dryness. The residue was then redissolved in an appropriate amount of 0.3 mol $L^{-1}$ $HNO_3$ for
concentration analysis, which was conducted using inductively coupled plasma quadrupole mass spectrometry (ICP-QMS,
Agilent 7700, Agilent, Japan). Acid digestion was conducted just once for each sample as the quantity of each sample was





limited. The average concentration of each element in the blank filter was subtracted from those in the samples. The Fe concentration in the blank filter was $2.5\pm0.6$ ngFe cm$^{-2}$ filter. Coarse particles were collected on the center part of the filter (5% of the filter area) whereas finer particles were collected on the outer part of the filter (95% of the filter area). Thus, the blank Fe concentration per the same volume of air was larger in fine particles than in coarse particles. Assuming that the total volume of filtered air was 720 m$^3$, the detection limits of Fe mass concentrations were approximately $1\times10^{-2}$ ng m$^{-3}$ and $2\times10^{-1}$ ng m$^{-3}$ for coarse and fine particles, respectively. Detection limits of the other elements were $4\times10^{-3}$ ng m$^{-3}$, $3\times10^{-5}$ ng m$^{-3}$, and $2\times10^{-4}$ ng m$^{-3}$ for coarse particles of Ti, V, and Pb, respectively, and $7\times10^{-2}$ ng m$^{-3}$, $5\times10^{-4}$ ng m$^{-3}$, and $3\times10^{-3}$ ng m$^{-3}$ for fine particles of Ti, V, and Pb, respectively.

The enrichment factor (EF), which is useful in understanding the impact of anthropogenic materials relative to crustal materials, was calculated according to the following equation:

$$\text{EF}_M = \frac{(M/Ti)_{sample}}{(M/Ti)_{crust}}, \tag{3}$$

where M is the concentration of the target element. The crustal value of each element was obtained from Taylor (1964).

### 2.3. Fractional Fe solubility

The fractional Fe solubility was calculated according to Eq. (1), for which soluble Fe (S-Fe) was evaluated via a leaching experiment. We applied the "flow-through extraction" method to solubilize the labile Fe component (Buck et al., 2006). For this method, extraction was conducted by pouring 100 mL of ultrapure water onto a sample filter in a pre-washed PTFE filter holder (47 mm single-stage filter assembly, Savillex, USA) equipped with a pre-washed 0.4 μm polycarbonate filter. A vacuum was applied during the extraction. The solution was collected in an LDPE vial and acidified with HNO$_3$ to avoid precipitation of the iron hydroxides, then evaporated in a PFA vial. The residue was then redissolved in 0.3 mol L$^{-1}$ HNO$_3$ for concentration analysis with ICP-QMS. The filter blank was $0.018\pm0.003$ ngFe cm$^{-1}$ filter, which was subtracted from each sample concentration. Although our previous studies applied different extraction methods, the flow-through method was applied in this study to compare the results with other studies of marine aerosols (e.g., Buck et al., 2006; Conway et al., 2019; Morton et al., 2013; Shelley et al., 2018).

### 2.4. Iron speciation

The average Fe species of each sample was determined by Fe K-edge X-ray absorption fine structure (XAFS) spectroscopy at the beamline BL-12C of the Photon Factory (PF), KEK (Ibaraki, Japan). The methods used were similar to those of previous studies (Kurisu et al., 2019; Takahashi et al., 2011). The X-ray was monochromatized using a Si(111) double-crystal monochromator and focused using a bent cylindrical mirror. The energy resolution was $\Delta E/E\sim0.2$ eV. The higher-order harmonic waves were eliminated using Ni-coated double mirrors. The reference spectra were recorded in transmission mode, whereas the sample spectra were recorded in fluorescence mode by placing the samples 45° from the incident beam. The Fe Kα (6.41 keV) line was detected using either a 19 element germanium solid-state detector or a 7 element silicon drift detector (SDD). X-ray absorption near-edge structure (XANES) spectra from 7.000 to 7.250 keV were recorded. Spectra were normalized at 7.200 keV and linear combination fitting was conducted in the energy range of 7.110 to 7.160 keV using REX2000 software (Rigaku Co., Ltd., Japan). For the energy calibration, the pre-edge peak energy of hematite was set to 7.112 keV. The quality of the fitting was checked using the parameter R, which is defined using

$$\sum R = \sum \{I_{obs}(E) - I_{cal}(E)\}^2 / \sum\{I_{obs}(E)\}^2, \tag{4}$$

where $I_{obs}(E)$ and $I_{cal}(E)$ are the X-ray absorbance of the original and calculated spectra, respectively, at a certain energy. The fitting errors for the fraction of the reference species were calculated as the difference in the fraction when the R-value is twice



the best-fitted value (Kodama et al., 2006). The error of each fitting was within 10 %. The samples contained just a small amount of Fe, so aerosol particles were mounted on a Kapton tape from the sampling filter to gather the particles at one point.

The micro-XAFS combined with X-ray fluorescence mapping (μ-XRF-XAFS) was conducted at BL-4A and BL-15A1 of the PF for analysis of the narrow spot speciation and to check the distribution of Fe and other elements. A 7.50 keV or 3.00 keV (for sulfur mapping) incident beam was used for elemental mapping. The Fe Kα (6.41 keV), potassium (K) Kα (3.31 keV), and sulfur (S) Kα (2.31 keV) lines were detected using a single element SDD. The incident beam sizes at the beamlines BL-4A and BL-15A were approximately $4\times5$ μm$^2$ and $20\times20$ μm$^2$, respectively. Note that the beam sizes were

larger than the fine particles (with aerodynamic diameters of < 2.5 μm); therefore, each spot could contain several particles. All of the aerosol sample spectra were recorded using the fluorescence mode of the SDD.

### 2.5. Iron isotope analysis

The sample preparation methods that were used for isotope analysis were the same as those described in Kurisu et al.

(2019) except for the method used to separate Fe with an anion exchange resin (AG-MP-1, 100–200 mesh, Bio-rad). The amount of the resin was reduced to lower the Fe contamination from the resin, which can be problematic for the analysis of small amounts of Fe, such as those found in marine aerosol samples. A microcolumn with a diameter of 2 mm was prepared with a heat-shrink PTFE tube (inner diameter: 7/8 inch, 4:1 heat shrink, Zeus Inc.), according to the method described by John and Adkins (2010). A frit was made of polyethylene. Approximately 20 μL of resin was used for each separation, with the

resin reaching a height of approximately 5 to 7 mm. The column was washed with 3.0 mol L$^{-1}$ HCl prior to use. The resin was washed by sequential soaking for one week in 0.5 mol L$^{-1}$ HNO$_3$ and 0.5 mol L$^{-1}$ HCl, respectively, with rinsing with ultrapure water between steps. The procedures that were used for the column separation are described in Table S3 and were modified from the method proposed by Conway et al. (2013). The procedure blank was 0.30±0.15 ngFe (1SD, n=3), which was more than 100 times lower than the amount of Fe in the sample. The recovery rate of Fe was 101±5 %, and the efficient removal of

other elements was confirmed (Fig. S1).

Iron isotope analysis was conducted using a multi-collector ICP-MS (MC-ICP-MS, Neptune Plus, Thermo Fisher Scientific, Germany). The methods used for the analysis were also the same as those described in Kurisu et al. (2016b, 2019). A standard-sample bracketing method was adopted, with Cu-doping as an external standard for mass bias correction using the exponential law (Albarede et al., 2004). The medium-resolution mode was used to resolve any argide interference (mainly

$^{40}Ar^{14}N^+$, $^{40}Ar^{16}O^+$, $^{40}Ar^{16}OH^+$, and $^{40}Ar^{18}O^+$). Data were obtained in dynamic mode with the Faraday cup setting to monitor the isotopes of $^{52}Cr$, $^{54}Fe$, $^{56}Fe$, $^{57}Fe$, and $^{58}Fe$ in one cycle and $^{63}Cu$, and $^{65}Cu$ in another. Iron isotope ratios were reported as the ratio of $^{56}Fe$ to $^{54}Fe$ according to Eq. (2). The ratio of $^{57}Fe$ to $^{54}Fe$ was used to check the reliability of the measurement in a three-isotope plot. The interference of $^{54}Cr$ with $^{54}Fe$ was less than 0.1 mol% in each sample and was corrected using $^{52}Cr/^{54}Cr=0.0282$ (Beard et al., 2003). By comparing the δ$^{56}$Fe in a Cr-free standard solution with the same solution under Cr-

doping ($^{54}Cr/^{54}Fe$ up to 0.01 in mol fraction), it was confirmed that no bias was observed between them. The Fe concentration was adjusted to 500 μg L$^{-1}$ for the isotope measurement. As the amount of Fe in each sample was small, the volume of the samples was low; therefore, limited iterations of the isotope measurements could be conducted.

Since the aerosol sampling filters contained some blank Fe, the δ$^{56}$Fe for each sample was corrected by measuring the δ$^{56}$Fe of a blank filter (δ$^{56}$Fe$_{blank}$, 0.13±0.07 ‰, n=3). Based on the blank Fe/total Fe ratios (Tables S4 and S5), the δ$^{56}$Fe

of the sample Fe (δ$^{56}$Fe$_{sample}$) was calculated according to the following mixing equation:

$$\delta^{56}Fe_{measured} = \delta^{56}Fe_{sample} \times (fraction\ of\ sample\ Fe) + \delta^{56}Fe_{blank} \times (fraction\ of\ blank\ Fe). \qquad (5)$$

Some samples were not available because the blank Fe was high (more than approximately 40 % of the total Fe) or because the amount of sample Fe was too small. In other samples, δ$^{56}$Fe$_{measured}$ and δ$^{56}$Fe$_{sample}$ were similar in their range of error (Tables S4 and S5); thus, this correction did not seriously affect the results and discussion of the isotope data. Although the Fe isotope





and speciation data of samples 14-N and 14-O have already been reported in Kurisu et al. (2016b), they were combined with the other newly obtained data in this study.

**2.6. Atmospheric chemical transport model**

This study compared the results of the fraction of combustion Fe in aerosols with those estimated by the Integrated
Massively Parallel Atmospheric Chemical Transport (IMPACT) model (Ito et al., 2021; Rotman et al., 2004). Simulations were performed for the period extending from 2013 to 2014, using a horizontal resolution of 2.0°×2.5° and 47 vertical layers. The IMPACT model categorized the Fe origins into mineral dust (Comp 1), oil combustion (ship emission, Comp 2), anthropogenic combustion on land (mainly coal combustion and the iron and steel industry, Comp 3), and biomass burning (Comp 4) for four distinct aerosol size bins (0.1–1.26 µm, 1.26–2.5 µm, 2.5–5 µm, and 5–20 µm in diameter). We compared
the sum of (Comp 2) and (Comp 3) with the isotope-based estimation for fine (i.e., the sum of bins 1 and 2) and coarse (i.e., the sum of bins 3 and 4) particles. Although biomass burning (Comp 4) may contain a combustion Fe source, we did not include it for the comparison. This is because the Fe emitted by biomass burning does not yield $\delta^{56}$Fe values that are as low as those from other anthropogenic combustion sources, which is likely due to strong influences by mineral dust aerosols entraining into the atmosphere during biomass burning (Kurisu and Takahashi, 2019).

**3. Results**

**3.1. Sources of aerosols**

A seven day backward trajectory analysis was conducted using the Hybrid Single-Particle Lagrangian Integrated Trajectory (HYSPLIT) model to discern the source of the air masses (Stein et al., 2015), which were then divided into three broad groups (Fig. 2). The first group was mainly transported from East Asia via westerly winds and corresponded to samples
13-a, 13-d, 13-e, 14-A, 14-N, and 14-O (group I, Fig. 2a). The second group was from the central and eastern Pacific and comprised samples 13-b, 13-c, 14-B, 14-C, 14-D, 14-E, 14-F, 14-G, 14-H, and 14-I (group II, Fig. 2b). The third group was from the northern North Pacific and was composed of samples 14-J, 14-K, 14-L, and 14-M (group III, Fig. 2c).

The Fe concentration was significantly affected by the source of air masses. The bulk Fe concentrations (fine+coarse) were 2.6 to 41.8 ng m$^{-3}$ in the vicinity of East Asia (group I), whereas the samples in groups II and III showed Fe concentrations
of less than 5 ng m$^{-3}$ (Fig. 3a, Tables S4 and S5). The concentrations were relatively low, even in group I, compared with concentrations reported from previous studies conducted in the Pacific (1 to 800 ng m$^{-3}$; Buck et al., 2006, 2019; Duce and Tindale, 1991; Sakata et al., 2018), in which more than 100 ng m$^{-3}$ of Fe was often observed during dust events from East Asia. Therefore, it is assumed that our samples were not affected by a large dust event from East Asia. Samples 14-G and 14-H showed Fe concentrations that were slightly higher than the samples from neighboring locations, especially in the coarse
particles, which may be derived from the Hawaiian islands. The Fe concentrations of the fine particles in the group III samples were below the detection limit, because of a smaller quantity of Fe in the samples than that in the blank.

The concentrations of other elements are also shown in Fig. S2. Titanium (Ti), which is abundant in the Earth's crust, showed similar concentration distributions to those of Fe (Fig. S2a). In addition, V and Pb, which originate mainly from oil and coal combustion (e.g., Nriagu and Pacnya, 1988; Zhang et al., 2009), respectively, were also found at high concentrations
near East Asia (group I, Figs. S2b and S2c) and were at especially high concentration in the fine particles, suggesting the significant influence of material from combustion sources. To further investigate the sources of the aerosols, the EF values were also compared. An EF value of more than 10 usually suggests a significant influence of anthropogenic materials (Buck et al., 2019; Shelley et al., 2015). The EF$_{Fe}$ values were 0.4 to 3 (Fig. 4a), although several of the samples showed large errors. There was no significant difference among the three groups. This result suggests that Fe mainly originated from crustal sources,

with no clear signal of anthropogenic Fe in the EF$_{Fe}$ results alone. EF$_{Pb}$ values were more than 10 for most of the samples, especially in the fine particles, indicating that most of the samples contained anthropogenic Pb (Fig. 4c). The EF$_{Pb}$ of the fine particles in group I was higher that of the other samples, especially in the winter (KH-13-7), which is indicative of a large influence from coal combustion in East Asia. EF$_V$ values varied from 1.7 to 840, with higher EF$_V$ values in fine particles than in coarse particles (Fig. 4b). The EF$_V$ values of the fine particles in the group III samples were more than 10, higher than the

other samples, which was not the case for Pb. Because V is mainly emitted during the combustion of heavy oil, the group III samples were likely influenced by ship emissions, which is plausible considering that many ships traverse the northern Pacific (https://www.marinetraffic.com).

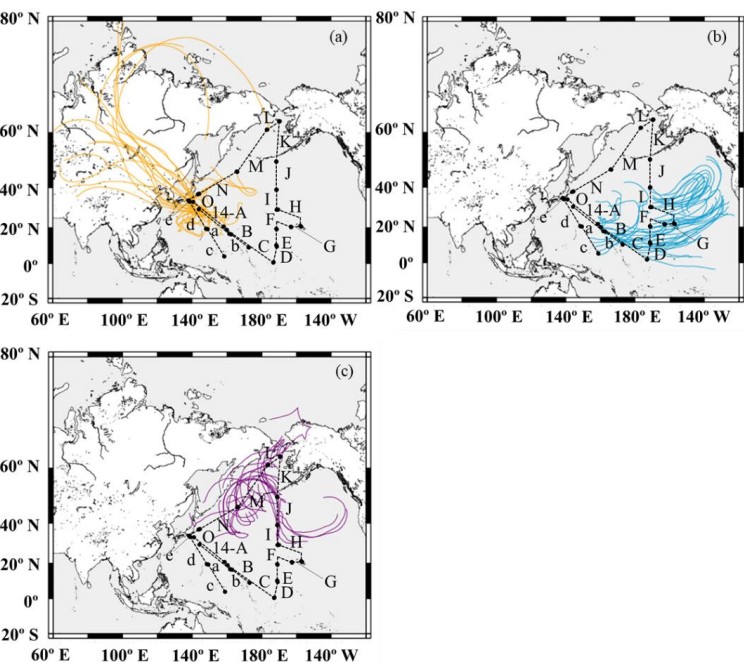

**Figure 2. Seven-day backward trajectories along the ship tracks for (a) group I, (b) group II, and (c) group III. The arrival height was set to 500 m and a new trajectory was obtained every 6 hours. The maps were created using GMT.**

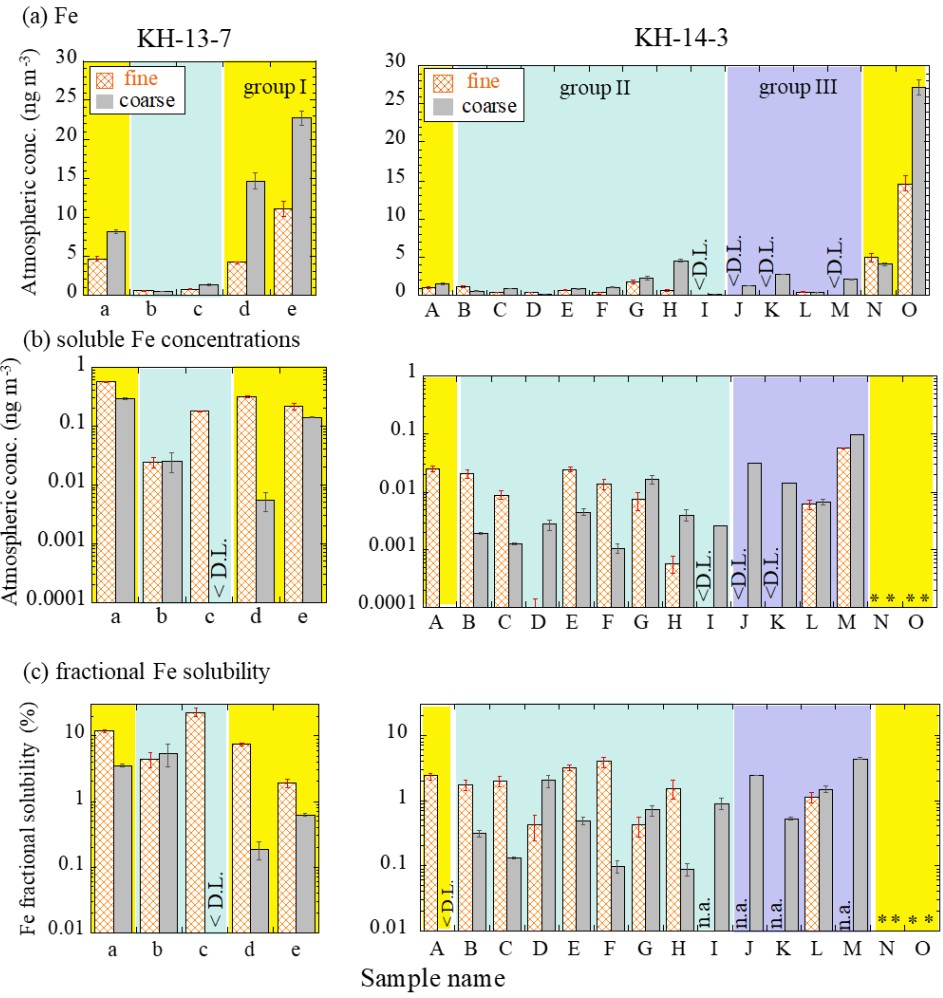

**Figure 3. Atmospheric concentrations of (a) Fe and (b) soluble Fe, and (c) fractional solubility of Fe of fine and coarse particles during the KH-13-7 and KH-14-3 cruises. Errors were calculated from ICP-MS errors and blank subtraction errors. Yellow, blue, and purple areas indicate the group I (air masses from the Asian continent), II (air masses from the central and eastern Pacific), and III (air masses from the northern North Pacific), respectively. <D.L., below the limit of detection limit due to higher concentrations in the blanks than samples; n.a., not available because the total Fe concentration was too low, * not analyzed due to insufficient sample amount was not enough.**

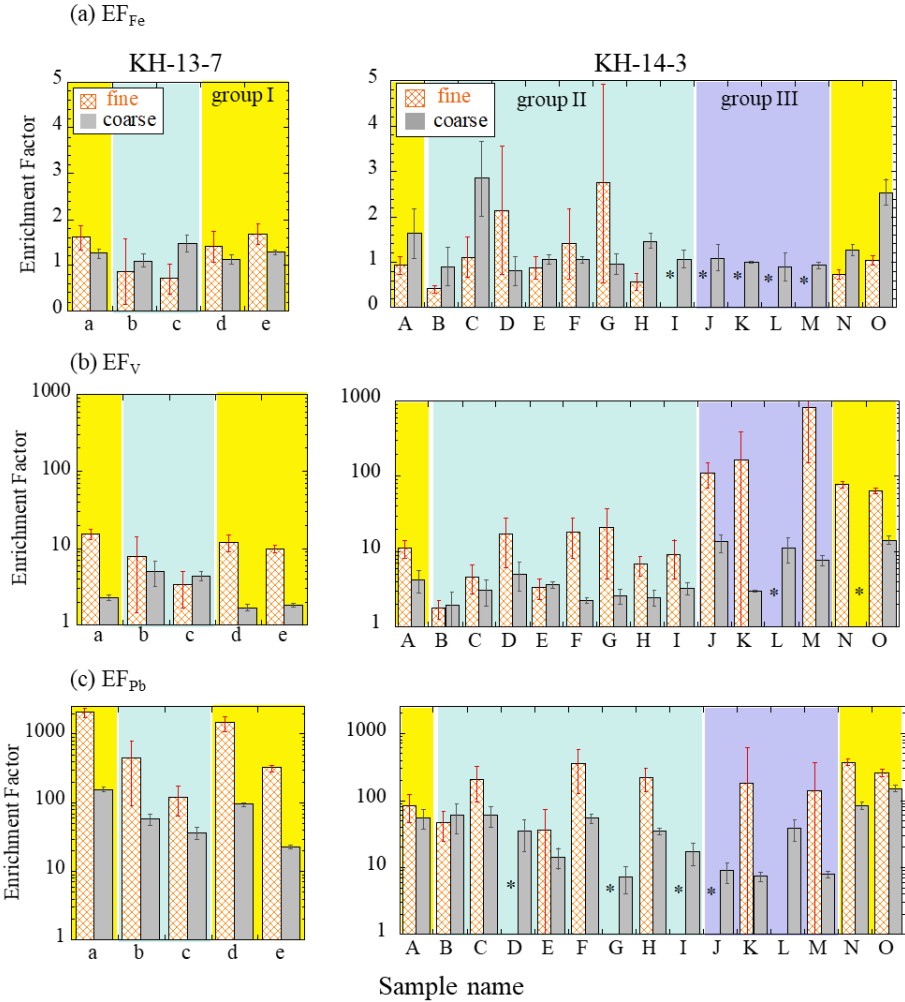

**Figure 4. Enrichment factors of (a) Fe, (b) V, and (c) Pb. Errors were calculated from ICP-MS error and blank subtraction errors. Yellow, blue, and purple areas indicate group Is (air masses from the Asian continent), II (air masses from the central and eastern Pacific), and III (air masses from the northern North Pacific), respectively. *Not available due to insufficient sample.**

### 3.2. Fractional Fe solubility

The fractional Fe solubility varied from 0.1 to 23 %, with the highest solubility observed in the fine particles in 13-c (Fig. 3c). The fractional solubilities were in a similar range to those observed in other studies conducted in the North Pacific (Buck et al., 2006; Wang and Ho, 2020; Wu et al., 2007). There were no clear differences in the fractional solubilities of the different air mass groups. Fine particles often showed higher soluble Fe concentration and fractional Fe solubility than coarse particles, which is consistent with previous reports (Kurisu et al., 2016b; Ooki et al., 2009). A scatter plot of the atmospheric Fe concentration and the fractional Fe solubility also showed a similar trend to previous studies, with high fractional solubilities obtained when Fe concentrations were low (Fig. S3; Mahowald et al., 2018; Sholkovitz et al., 2009, 2012); the reason for this trend will be discussed later.



### 3.3. Iron species

The difference in the peak energies of the XANES spectra reflects the oxidation state of Fe. The higher peak energy of the Fe K-edge XANES spectrum corresponds to a larger amount of ferric species, as seen in the reference materials (Fig. 5a; Berry et al., 2003; Maggi et al., 2018). The peak energies of some of the samples (such as the coarse particles of sample 13-a) were approximately 7.126 keV, whereas those of other samples (such as the fine particles of 13-a) were higher in the range between 7.126 and 7.131 keV, indicating that they contained a larger fraction of ferric species. The spectra of the samples were successfully fitted by linear combination to those described by the reference samples. The ratios obtained by fitting are tabulated in Table S6. For simplification, the main Fe species were categorized into Fe-containing silicates (weathered biotite, chlorite, and illite with various Fe(II) fractions), Fe oxides (magnetite, goethite, and hematite), and Fe(III) hydroxides (ferrihydrite, Fig. 5e), considering that Fe-containing silicates are of natural origin, whereas Fe (hydr)oxides can be both of natural and combustion origin. We distinguished between Fe oxides and Fe(III) hydroxides because the latter have higher fractional Fe solubility and are often formed by reacting with acids during atmospheric transportation (Shi et al., 2011; Takahashi et al., 2011). Most of the sample spectra were fitted with Fe-containing silicates and ferrihydrite. Fe oxides were only found in the group III samples from the northern North Pacific (samples 14-J to 14-M), suggesting the presence of Fe particles from different sources. The fraction of each species in coarse and fine particles was similar within a range of approximately 10 % error in most of the group II samples from the central and eastern Pacific, whereas the fractions of ferrihydrite in the group I samples from East Asia and some of the group II samples were higher in fine particles than in coarse particles.

The presence of Fe-containing silicates was confirmed by spot analysis of some of the samples using µ-XRF-XAFS. For example, there were several Fe-concentrated spots in the coarse and fine particles (Figs. S4−S7). These were Fe-containing aluminosilicates, such as weathered biotite, illite, and chlorite, the presence of which was also supported by the correlation with the intensity of potassium (K) in the µ-XRF maps (Figs. S4b and S4d). These species were also previously observed in aerosols that were collected in East Asia (Kurisu et al., 2016b; Takahashi et al., 2011), suggesting the presence of mineral dust from East Asia. The presence of Fe (hydr)oxides was also observed in the spot analysis, with some particles containing ferrihydrite, goethite, magnetite, and hematite (Figs. S5, S6, and S7). Ferrihydrite was often found in the fine particles within sample 13-c that was collected in the open ocean, which was also consistent with the bulk XANES analysis (Fig. S6). In addition to Fe-containing aluminosilicates and Fe (hydr)oxides, Fe(III) sulfate, a highly soluble Fe species, was also observed in the coarse and fine particles of sample 13-c, although it was not detected as main Fe species in the bulk analysis. The presence of Fe(III) sulfate was supported by the co-presence of Fe and S (spots 5 and 6 in Fig. S5b).

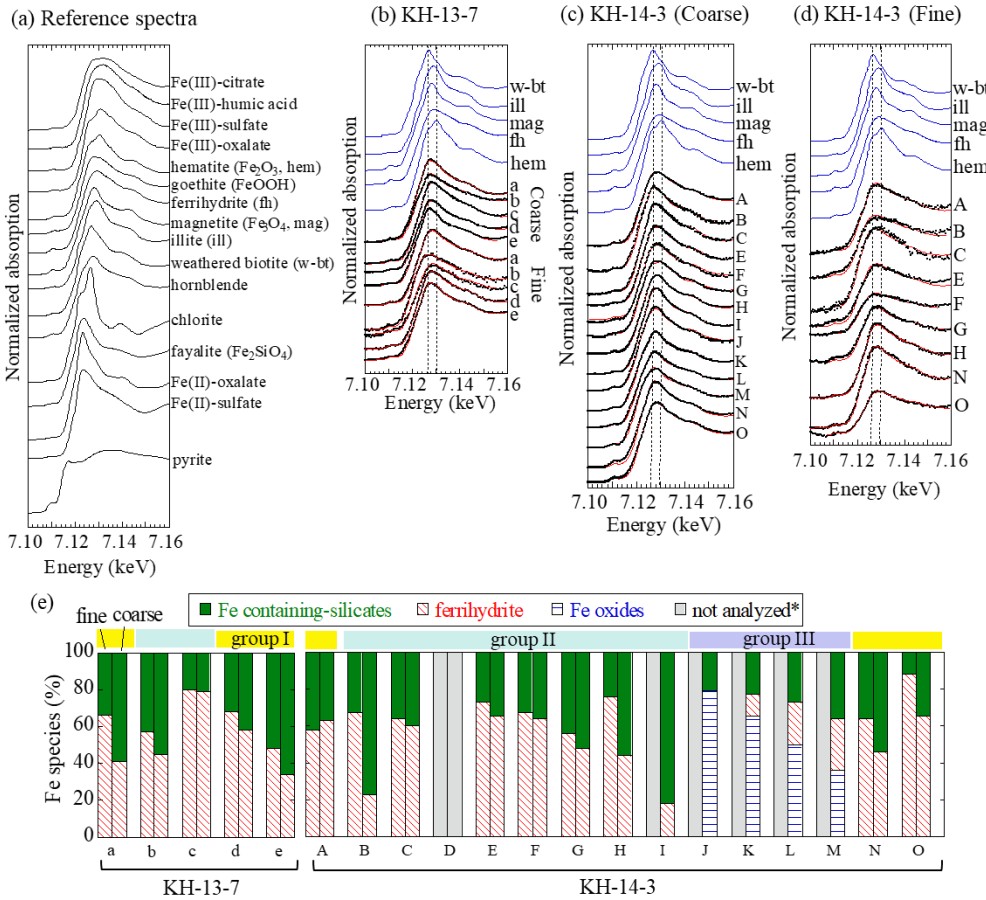

**Figure 5.** Iron K-edge XANES spectra of (a) reference species, (b) KH-13-7 samples, (c) coarse particles of KH-14-3 samples, and (d) fine particles of KH-14-3 samples. (e) Fraction of each Fe species in the fine and coarse samples. Colored bars above the graph (d) indicate the different air mass groups. *Not analyzed due to insufficient Fe. Yellow, blue, and purple areas indicate groups I (air masses from the Asian continent), II (air masses from the central and eastern Pacific), and III (air masses from the northern North Pacific), respectively.

### 3.4. Iron isotope ratios

Negative $\delta^{56}Fe$ values ranging from −0.45 to −2.16 ‰ were observed in the fine particles of group I, in air masses from East Asia (Fig. 6, Tables S4 and S5). These values were much lower than those observed for the coarse particles (on average 0.10 ‰). The $\delta^{56}Fe$ values were particularly low in the fine particles of samples collected in the vicinity of the Japanese coastline (13-a, 14-N, and 14-O). The bulk (coarse + fine) $\delta^{56}Fe$ values calculated from the Fe concentration and $\delta^{56}Fe$ of each size fraction ranged from −0.07 to −0.91 ‰ (Table S4), which is considerably lower than the values obtained from North American or European air masses in the North Atlantic, which reached as low as −0.16 ‰ (Conway et al., 2019), suggesting an importance of aerosols with low $\delta^{56}Fe$ values in the North Pacific. These low $\delta^{56}Fe$ values may have originated from combustion Fe, which is discussed in more detail later.





In air masses from the central, eastern, or northern Pacific, the $\delta^{56}$Fe values of both the coarse and fine particles in groups II and III were close to or higher than 0 ‰. The coarse and fine particles yielded similar $\delta^{56}$Fe values to each other,

although the $\delta^{56}$Fe of the fine particles in some samples could not be measured due to the low quantity of sample Fe compared with blank Fe. Most of the values were close to the crustal value, whereas the $\delta^{56}$Fe of some samples was higher than the crustal average, reaching as high as 0.43 ‰, which did not correspond with typical crustal values (−0.3 to 0.3 ‰, Johnson et al., 2003; Majestic et al., 2009; Mead et al., 2013).

The coarse and fine particles in sample 13-e and the fine particles of sample 13-d contained sufficient soluble Fe for

isotope analysis to be conducted (Table S4). The resulting values were −0.27±0.03 ‰, −1.14±0.03 ‰, and −2.23±0.04 ‰ for the coarse particles in 13-e, fine particles in 13-e, and fine particles in 13-d, respectively, which were lower than those of the acid-digested (total) Fe (0.10±0.14 ‰, −0.47±0.18 ‰, and −0.45±0.28 ‰, respectively), especially for fine particles (Table S4). These low values in fine particles were due to the preferential dissolution of components with a low $\delta^{56}$Fe value (i.e., combustion Fe). It should be noted that these lower $\delta^{56}$Fe values could be due to kinetic isotope fractionation taking place

during the partial dissolution of a single phase (Revels et al., 2015; Wiederhold et al., 2006). However, such fractionation would become large as the fractional Fe solubility decreases, which was not the case with our results; the difference between the $\delta^{56}$Fe values in the total and soluble fractions of the samples was greater when the fractional Fe solubility was large (Table S4 and Fig. 3c). It is possible that the soluble component of the coarse particles of 13-e had $\delta^{56}$Fe lower than that of the total due to kinetic isotope fractionation. However, the low $\delta^{56}$Fe values, especially in terms of the fine particles, cannot be explained

sorely by kinetic isotope fractionation and the component with a low $\delta^{56}$Fe value (combustion Fe) must have dissolved preferentially. The bulk (coarse + fine) $\delta^{56}$Fe value of −0.79 ‰ was calculated for 13-e from the concentration and $\delta^{56}$Fe of soluble Fe in each size component, and although the $\delta^{56}$Fe of the coarse particles in 13-d was not measured, the bulk (coarse + fine) $\delta^{56}$Fe value was probably close to the $\delta^{56}$Fe of the fine particles (−2.23 ‰), as most of the soluble Fe in sample 13-d was from fine particles (Fig. 3b and Table S4). These calculated bulk $\delta^{56}$Fe values were in a similar range to or lower than

those observed previously in the North Atlantic (reaching as low as −1.6 ‰, Conway et al., 2019), indicating the significance of low-$\delta^{56}$Fe components (i.e. combustion Fe) in the northwestern Pacific.

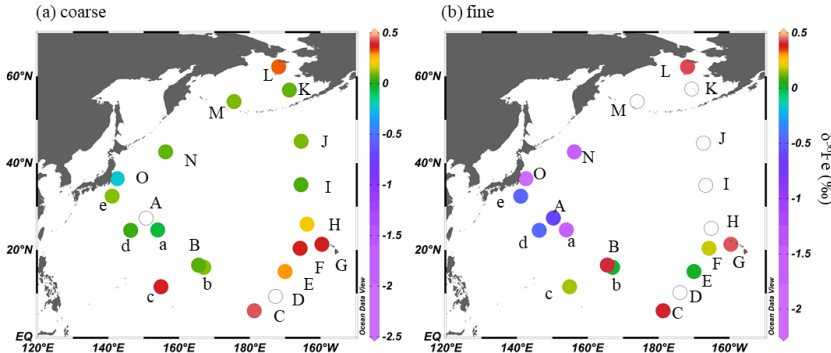

**Figure 6. Iron isotope ratios of KH-13-7 and KH-14-3 samples in (a) coarse and (b) fine particles. Data in the clear circles were not available. Values are also shown in Tables S5 and S6. The figures were produced using Ocean Data View (Schlitzer, 2020).**




## 4. Discussion

### 4.1. Source apportionment of Fe in aerosols based on $\delta^{56}$Fe values

#### 4.1.1. Do the negative $\delta^{56}$Fe values in fine particles mean they are of combustion origin?

The Fe isotope analysis indicated that low $\delta^{56}$Fe values were observed in the fine particles within air masses from
East Asia. To check if the low $\delta^{56}$Fe values were a result of the presence of Fe from anthropogenic combustion, the relationship between $\delta^{56}$Fe and the concentration of anthropogenic constituents was explored (Fig. 7). The $\delta^{56}$Fe values were weakly correlated with $EF_{Pb}$, $EF_{V}$, and the concentration of carbon monoxide, which indicated that aerosols that contained anthropogenic materials also contained Fe with low $\delta^{56}$Fe values. Therefore, the low $\delta^{56}$Fe values observed in the aerosol samples were due to the presence of anthropogenic combustion Fe.

Other possible sources of the observed low $\delta^{56}$Fe values, such as biomass burning and volcanic emissions, were taken into account. There was an impact from biomass burning in Siberian forest fires during the sampling period of 14-M and 14-N, according to Jung et al. (2016) and Kamezaki et al. (2019). In addition, the fine particles in 14-N yielded a low $\delta^{56}$Fe value. However, our previous study suggested that the $\delta^{56}$Fe of aerosols emitted from biomass burning is not particularly low because of (i) the preferential emission of Fe by soil suspensions with a value of $\delta^{56}$Fe that is close to the crustal average and (ii) the
low combustion temperatures (300 to 500 °C), which are not high enough for the emission of large amounts of Fe to occur via evaporation (Kurisu and Takahashi, 2019). Although some plants have values of $\delta^{56}$Fe as low as −1.6 ‰ (Guelke and Von Blanckenburg, 2007), they are not considered to be an important Fe source emitted by biomass burning because the Fe content in plants is low compared with that of soil. The low $\delta^{56}$Fe value observed in 14-N may have been a result of the significant influence of combustion Fe originating from coal combustion, because a high $EF_{Pb}$ was observed (Nriagu and Pacnya, 1988;
Zhang et al., 2009). Taken together, these observations suggest that biomass burning cannot explain the low $\delta^{56}$Fe, although Fe from biomass burning may have influenced the results in terms of Fe concentrations.

There are no data concerning the $\delta^{56}$Fe of aerosols emitted by volcanic activity. The typical $\delta^{56}$Fe range of volcanic rocks is −0.3 to 0.3 ‰ (Fantle and DePaolo, 2004; Heimann et al., 2008), whereas it is possible that volcanic emissions have a low $\delta^{56}$Fe value as a result of isotope fractionation during evaporation, considering that Fe evaporates as gaseous Fe chloride
($FeCl_2$ or $FeCl_3$), based on thermodynamic calculations (Symonds et al., 1992); further studies are required to investigate this. According to the volcanic eruption database (Global Volcanism Program, Smithsonian Institution, https://volcano.si.edu/), there were several volcanic activity events around the Aleutian Islands during the sampling periods. However, low $\delta^{56}$Fe values were not observed around this area, suggesting that volcanic emission was not the reason for the low $\delta^{56}$Fe values observed in the present study.

To the best of our knowledge, there are no reported aerosol sources with a low $\delta^{56}$Fe other than anthropogenic combustion Fe. Thus, we conclude that the low $\delta^{56}$Fe values were due to the presence of combustion Fe.

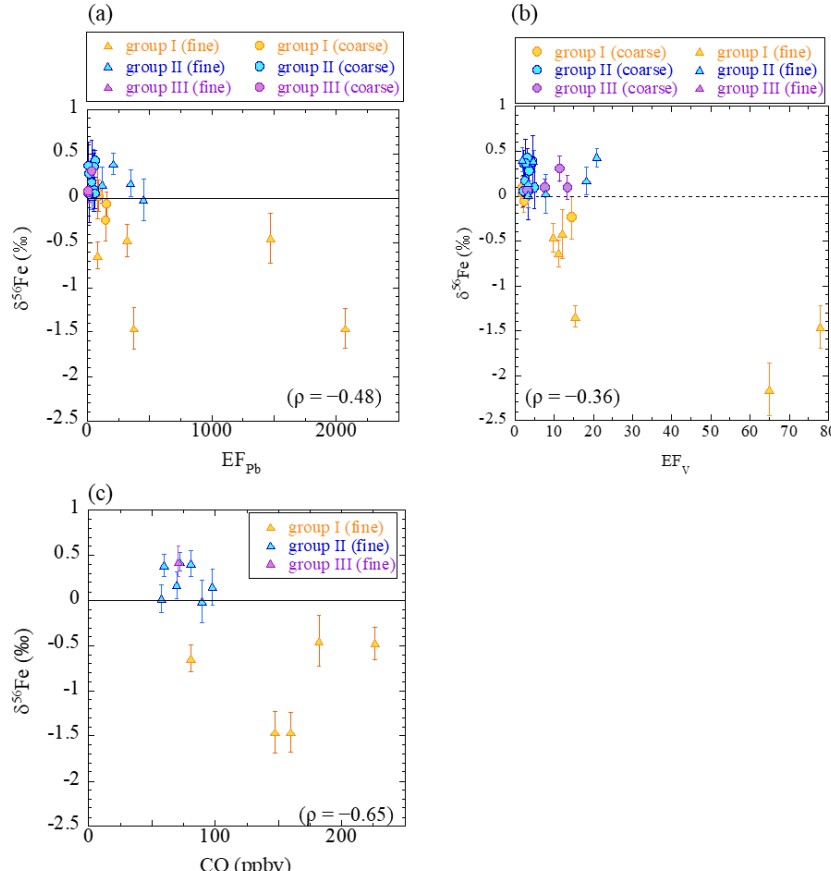

**Figure 7. Scatter plots of Fe isotope ratios and (a) EF$_{Pb}$, (b) EF$_V$, and (c) CO. Concentrations of CO were averaged values during the sampling period (data from Kamezaki et al., 2019). ρ is the Spearman Rank Order correlation coefficient (p<0.05). Yellow, group I; blue, group II; purple, group III.**

### 4.1.2. The origin of Fe with δ⁵⁶Fe values close to or higher than 0 ‰

Most of the coarse particles and some of the fine particles from samples collected over the open ocean yielded δ⁵⁶Fe values close to 0.0 ‰, the typical value of crustal materials (Beard et al., 2003; Johnson et al., 2003). Considering that the major source of Fe is mineral dust, most of the particles can be considered to be of crustal origin.

δ⁵⁶Fe values as high as 0.45 ‰ were observed in some of the samples, especially those collected over the open ocean. We confirmed that these values were not caused by interference from any other elements during measurements, such as ⁵⁴Cr on ⁵⁴Fe, or ⁴⁰Ar²³Na on ⁶³Cu. On the whole, high δ⁵⁶Fe values tended to be observed when the Fe concentration was low (< 5 ng m⁻³, Fig. S8a). Labatut et al. (2014) also reported that δ⁵⁶Fe values (0.27±0.15 ‰ to 0.38±0.08 ‰) in the western equatorial Pacific were relatively high compared with those of typical mineral aerosols. Possible explanations for these high δ⁵⁶Fe values are (i) another Fe source with high δ⁵⁶Fe values or (ii) the long-range transportation of lithogenic aerosols along with a change in δ⁵⁶Fe values.

In the case of (i), several sources can be considered, such as mineral dust with high δ⁵⁶Fe values and sea spray aerosols (SSA). The δ⁵⁶Fe values of crustal materials typically lie within the range of −0.3 to 0.3 ‰ (Fantle and DePaolo, 2004; Johnson



et al., 2003; Majestic et al., 2009; Mead et al., 2013), and an average $\delta^{56}$Fe value of 0.09±0.03 ‰ has been reported for the Chinese Loess Plateau (Gong et al., 2017). However, Fe isotope fractionation can occur during continental weathering, the extent of which is determined by the combination of equilibrium and kinetic processes (Cornell and Schwertmann, 2003;

Dideriksen et al., 2010; Liu et al., 2014; Wiederhold et al., 2007). If such secondary minerals are selectively transported as aerosols, $\delta^{56}$Fe values that are different from the average dust values may be observed. However, such high $\delta^{56}$Fe values have not been reported in coarse aerosols on land, which contain a larger fraction of mineral dust. Therefore, the reason why only marine aerosols show such high $\delta^{56}$Fe values is unclear. Surface seawater generally has a high $\delta^{56}$Fe of dissolved Fe, up to 0.7 ‰, as a result of biological uptake of lighter Fe (Conway and John, 2014; Ellwood et al., 2015; Radic et al., 2011). The

concentration of dissolved Fe in the surface seawater of the open ocean in the North Pacific is generally low, typically less than 0.2 nmol L$^{-1}$ (Nishioka and Obata, 2017), whereas its concentration is approximately 2 orders of magnitude higher in SSA compared with its concentration in the surface seawater, according to Weisel et al. (1984). Although no data are available concerning the $\delta^{56}$Fe of SSA, it is possible that high values of $\delta^{56}$Fe are reflected in SSA. Considering that SSA contain Na as the main component, the Fe/Na molar ratios and $\delta^{56}$Fe values were compared (Fig. S9b). However, no clear trend was obtained,

suggesting that the high $\delta^{56}$Fe values cannot be explained by the presence of SSA alone.

In the case of explanation (ii), in which the long-range transportation of lithogenic aerosols is accompanied by a change in $\delta^{56}$Fe values, a proportion of each particle with a low $\delta^{56}$Fe value needs to be selectively removed to change the aerosol $\delta^{56}$Fe values during transportation. For example, considering the photochemical or acidification process of a single particle, the surface of the particle preferentially reacts with (acidic) water and becomes more soluble, isotope fractionation

occurs between the soluble and insoluble phases, and the soluble and insoluble phases become enriched with lighter and heavier Fe isotopes, respectively (Kiczka et al., 2010; Revels et al., 2015; Wiederhold et al., 2006). However, the selective removal of the soluble phase of the particle is needed to change aerosol $\delta^{56}$Fe values. Although soluble Fe is preferentially incorporated into clouds and removed by precipitation, it is unknown whether the selective removal of the soluble phase in a single particle occurs. In addition, coarse particles are not often incorporated into clouds. This possibility, therefore, requires further study.

Although it is not easy to obtain any clear conclusions at present, clarifying the reason for the high $\delta^{56}$Fe will lead to a better understanding of the transportation processes and sources of Fe. We plan to investigate this in the future, as the current study was focused on whether the low $\delta^{56}$Fe observed is caused by combustion processes.

### 4.2. Factors controlling fractional Fe solubility

As discussed in section 4.1, low $\delta^{56}$Fe values suggest the presence of combustion Fe. By comparing the $\delta^{56}$Fe values

of total (acid-digested) Fe and fractional Fe solubilities, the importance of combustion Fe as a controlling factor to determine fractional Fe solubility can be discussed. A good correlation between the fractional Fe solubility and $\delta^{56}$Fe value of the total Fe was observed in the group I samples (Fig. 8a). Assuming that combustion Fe has a constant $\delta^{56}$Fe value, a lower $\delta^{56}$Fe value corresponds to a larger fraction of combustion Fe in a sample. The fact that the soluble fraction in the fine particles of group I yielded $\delta^{56}$Fe values lower than the total (acid-digested) Fe (Table S4) indicated that combustion Fe with a low $\delta^{56}$Fe value

was preferentially extracted.

Previous studies have suggested that combustion Fe is more soluble than natural Fe because it is present mainly as an amorphous glass or aggregates of iron (hydr)oxide nanoparticles, which have higher solubilities and dissolution rates than the natural Fe that is usually found in aluminosilicate structures (Chen et al., 2012; Ito and Shi, 2016). Furthermore, combustion Fe is often emitted along with other acids, such as sulfates, leading to greater aerosol acidity and thus larger fractional Fe

solubility (Buck et al., 2006; Ingall et al., 2018; Li et al., 2017). Our results indicated a weak correlation between the fraction of (hydr)oxides (in this case ferrihydrite was the main Fe species) and the $\delta^{56}$Fe value (Fig. 8c) and between the fraction of (hydr)oxides and the fractional Fe solubility in the group I samples, although there were insufficient data concerning the latter (n=4, Fig. 8e). These results indicated that combustion Fe contained a large fraction of ferrihydrite, which was partially





responsible for the high fractional Fe solubilities. Therefore, the presence of combustion Fe was an important factor in

controlling the fractional solubility in the area from which the group I samples were taken.

The fractional Fe solubilities of the group II and III samples were not correlated with $\delta^{56}$Fe (Fig. 8b); although their $\delta^{56}$Fe values were close to or higher than 0 ‰, their fractional Fe solubilities varied from 0.1 to 23 %. Therefore, factors other than the effect of combustion Fe may be more important for controlling the fractional Fe solubility in the open ocean. Although some of the fine particles showed higher fractional Fe solubilities when the fraction of ferrihydrite was high, this was not the

case for the other samples (Fig. 8d), suggesting that the presence of ferrihydrite cannot fully explain the variation in the fractional Fe solubilities. One possibility is the presence of other Fe species with high fractional Fe solubilities that could not be detected by XAFS analysis (due to smaller fractions of < 10 %). For example, Fe sulfate has a high fractional Fe solubility, which was only found by μ-XAFS analysis (Fig. S5). In addition, other factors, such as in situ aerosol acidity or the presence of Fe-binding organic ligands (e.g., oxalate) may be important for Fe dissolution following long-range atmospheric

transportation (Chen and Grassian, 2013; Ito and Shi, 2016).

As noted in section 3.2, high fractional solubilities were obtained when Fe concentrations were low (Fig. S3). This trend is considered to be caused by (i) the longer lifetime of fine particles that contain large amounts of combustion Fe with high fractional solubility and/or (ii) an increase in the fractional Fe solubility during atmospheric transportation (Mahowald et al., 2018). From a scatter plot showing the atmospheric Fe concentration and fractional Fe solubility in which the color scale

shows the difference in the values of $\delta^{56}$Fe (Fig. S9), the importance of the presence of combustion Fe can be investigated. A high fractional Fe solubility was not obtained only when the $\delta^{56}$Fe was low, again suggesting that a combination of several factors controls fractional Fe solubility. The processes that influence fractional Fe solubility could be investigated using the Fe isotope ratio as another parameter.

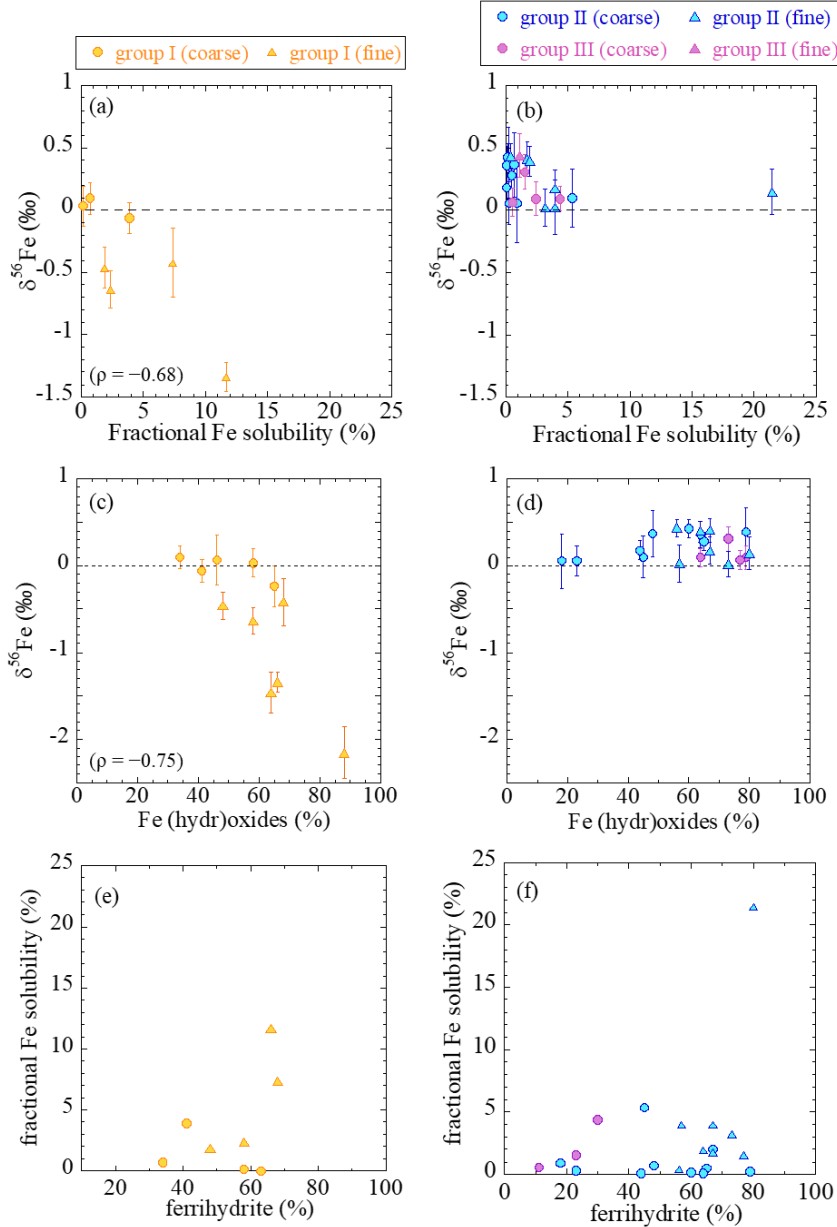


**Figure 8.** The relationships between (a,b) the fractional Fe solubility and $\delta^{56}$Fe; (c, d) the fraction of Fe (hydr)oxides and $\delta^{56}$Fe; and (e, f) the fraction of Fe (hydr)oxides and the fractional Fe solubility. ρ is the Spearman Rank Order correlation coefficient (p<0.05). *Only for fine particles. Yellow, group I; blue, group II; purple, group III.


### 4.3. The contribution of combustion Fe to marine aerosols

We estimated the relative contribution of combustion Fe based on the mass balance equation for coarse (> 2.5 μm) and fine (< 2.5 μm) particles using the following equations:




$$\delta^{56}\text{Fe}_{fine} = \delta^{56}\text{Fe}_{combustion} \times f_{combustion-fine} + \delta^{56}\text{Fe}_{natural} \times (1 - f_{combustion-fine}),\qquad(6)$$

$$\delta^{56}\text{Fe}_{coarse} = \delta^{56}\text{Fe}_{combustion} \times f_{combustion-coarse} + \delta^{56}\text{Fe}_{natural} \times (1 - f_{combustion-coarse}),\qquad(7)$$

where $f_{combustion-fine}$ and $f_{combustion-coarse}$ are the fractions of combustion Fe in fine and coarse particles, respectively. The representative $\delta^{56}\text{Fe}$ value of combustion Fe was estimated to be −3.9 to −4.7 ‰ based on the $\delta^{56}\text{Fe}$ values of the aerosol samples collected at various sites, which included suburban areas and sites near to sources of anthropogenic emissions (Kurisu

et al., 2016a, 2016b, 2019; Kurisu and Takahashi, 2019). The $\delta^{56}\text{Fe}$ value of natural Fe was assumed to be the same as the crustal average of 0.0 ‰ (Beard et al., 2003). We assumed that samples were of natural origin even when they exhibited values of $\delta^{56}\text{Fe}$ higher than 0.0 ‰ based on the discussion in section 4.1. Although the evaluation of samples 14-N and 14-O has been already discussed in Kurisu et al. (2016b, 2019), in the present study these results were combined with the data describing the other 18 samples.

The calculated fractions of combustion Fe in fine and coarse particles were up to 50 % and 6 %, respectively, near the Japanese coast, and up to 21 % when calculating the fraction for the bulk (fine + coarse) particles. These results suggest the importance of combustion Fe even in marine aerosols (Fig. 9, Table 1). The fractions in the group I samples were higher in the summer (KH-14-3, on average 20 %) than in the winter (KH-13-7, on average 6 %), which was reasonable considering that the influence of mineral dust from East Asia is often greater during the winter. However, the influence of combustion Fe

was only seen in group I (approximately 2000–3000 km from East Asia) and was not observed in the group II and III regions. Natural Fe was more important when air masses were from the eastern, central, or northern Pacific.

The fraction of combustion Fe to soluble Fe was also calculated from the $\delta^{56}\text{Fe}$ of the soluble component in the sample 13-e (−1.14±0.03 ‰, Table S4). We assumed that the $\delta^{56}\text{Fe}$ of the soluble component of combustion Fe is the same as the total fraction (−3.9 to −4.7 ‰) and that the $\delta^{56}\text{Fe}$ of the soluble component of natural Fe is the same as that of the coarse particles

in 13-e (−0.27±0.03 ‰, Table S4), considering that kinetic isotope fractionation of natural Fe also occurred in the fine particles as discussed in 3.4. The ratio of combustion Fe in the soluble Fe of fine particles was approximately 22±2 %, which was higher than that of the total (soluble + insoluble) Fe (11±4 %, Table 1). From these results, the fractional Fe solubilities of natural and combustion Fe could also be calculated according to the following equation:

$$fractional\ Fe\ solubility_{comb}\ (\%) = \left(S-Fe_{fine} \times f_{sol-comb-fine}\right)/\left(T-Fe_{fine} \times f_{total-comb-fine}\right) \times 100,\quad(8)$$

where S-Fe$_{fine}$ and T-Fe$_{fine}$ are the soluble and total (acid-digested) Fe concentrations, respectively, and $f_{sol-comb-fine}$ and $f_{total-comb-fine}$ are the fractions of combustion Fe in soluble and total fractions, respectively, estimated by Eq. (6). The same calculation was conducted for natural Fe. The calculated fractional Fe solubilities were 0.9 % and 11 % for natural and combustion Fe, respectively. The higher fractional Fe solubility of combustion Fe than natural Fe again suggests the importance of combustion Fe as a source of soluble Fe, even though the emissions of combustion Fe are much smaller than natural Fe emissions.




Atmospheric
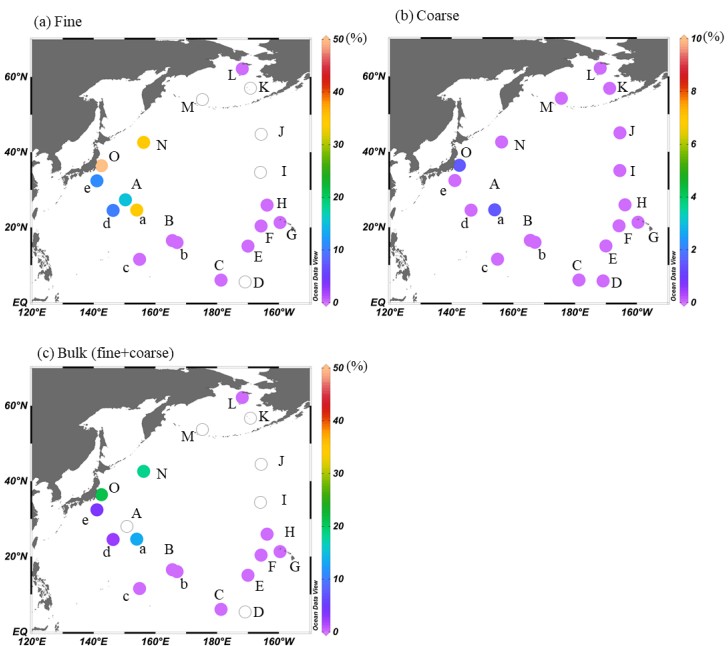

**Figure 9.** The contribution of combustion Fe in (a) fine, (b) coarse, and (c) total (fine + coarse) Fe in marine aerosols. Data with open circles were not determined. The figures were produced using Ocean Data View (Schlitzer, 2020).

**Table 1.** Comparison of the contribution of combustion Fe in total Fe estimated by the Fe isotope ratio and the IMPACT model. Yellow: group I, light blue: group II, purple: group III; * Errors were calculated based on 2 SD of the $\delta^{56}$Fe of samples and combustion Fe.

| Sample | Contribution of combustion Fe in total Fe in aerosols (%) | | | | | | | | |
|---|---|---|---|---|---|---|---|---|---|
| | Fine | | | Coarse | | | Fine + Coarse | | |
| | Isotope* | IMPACT model | | Isotope | IMPACT model | | Isotope | IMPACT model | |
| | | (ii) Anthropogenic combustion on land (mainly coal combustion) | (iii) Oil combustion | | (ii) Anthropogenic combustion on land (mainly coal combustion) | (iii) Oil combustion | | (ii) Anthropogenic combustion on land (mainly coal combustion) | (iii) Oil combustion |
| 13-a | 34±6 | 36 | 1 | 2±4 | 1 | 0 | 13±3 | 34 | 1 |
| 13-b | 0 | 17 | 1 | 0 | 0 | 1 | 0 | 16 | 1 |
| 13-c | 0 | 3 | 0 | 0 | 0 | 0 | 0 | 3 | 0 |
| 13-d | 10±7 | 21 | 1 | 0 | 0 | 0 | 2±1 | 17 | 1 |
| 13-e | 11±4 | 26 | 1 | 0 | 39 | 0 | 4±1 | 32 | 1 |
| | | | | | | | | | |
| 14-A | 15±5 | 4 | 7 | n.a. | 0 | 3 | n.a. | 4 | 7 |
| 14-B | 0 | 1 | 1 | 0 | 0 | 5 | 0 | 1 | 1 |
| 14-C | 0 | 1 | 2 | 0 | 0 | 2 | 0 | 1 | 2 |
| 14-D | n.a. | 0 | 2 | n.a. | 0 | 10 | n.a. | 0 | 2 |
| 14-E | 0 | 0 | 1 | 0 | 0 | 0 | 0 | 0 | 1 |
| 14-F | 0 | 1 | 1 | 0 | 1 | 1 | 0 | 1 | 1 |
| 14-G | 0 | 1 | 1 | 0 | 31 | 2 | 0 | 1 | 1 |
| 14-H | 0 | 2 | 3 | 0 | 17 | 2 | 0 | 3 | 3 |
| 14-I | n.a. | 1 | 3 | 0 | 0 | 14 | n.a. | 1 | 3 |
| 14-J | n.a. | 6 | 16 | 0 | 0 | 0 | n.a. | 6 | 14 |
| 14-K | n.a. | 6 | 12 | 0 | 0 | 1 | n.a. | 5 | 10 |
| 14-L | 0 | 4 | 8 | 0 | 1 | 1 | 0 | 4 | 7 |
| 14-M | n.a. | 3 | 9 | 0 | 0 | 0 | n.a. | 3 | 8 |
| 14-N | 34±6 | 9 | 7 | 0 | 1 | 0 | 18±4 | 8 | 6 |
| 14-O | 50±8 | 45 | 5 | 6±6 | 97 | 0 | 21±5 | 65 | 3 |





**4.4. Comparison with a model estimation**

The contributions of combustion Fe to the *total* (soluble + insoluble) Fe in the marine aerosols estimated by the IMPACT model during the sampling periods were compared with those estimated by the Fe isotope ratios (Table 1).

The contributions from combustion Fe estimated by the Fe isotope ratios and the IMPACT model were similar in fine particles of groups I and II (Table 1). The high proportions of combustion Fe in the fine particles in group I were reproduced

mainly by (Comp 3) anthropogenic combustion on land (mainly coal combustion) in the IMPACT model, whereas there were smaller contributions from oil combustion. The low contributions of combustion Fe in group II were also reproduced by the IMPACT model. Although only the $\delta^{56}$Fe of fine particles of sample 14-L could be analyzed in group III, the IMPACT model estimated relatively large contributions from combustion Fe (mainly oil combustion) in group III, which was not suggested by the Fe isotope ratio. It is possible that (i) the IMPACT model overestimated the contribution of Fe from oil combustion in

remote oceans or (ii) the $\delta^{56}$Fe value of combustion Fe estimated at −3.9 to −4.7 ‰ could not be applied to the estimation of the contribution of oil combustion. Because most Fe in liquid fuels might be released into the atmosphere (e.g., 93% according to Wang et al., 2003), there might be limited Fe isotope fractionation during oil combustion. As there are no data concerning the $\delta^{56}$Fe of atmospheric aerosols emitted by the combustion of oil, further studies regarding the $\delta^{56}$Fe value of a more specific source are required.

The combustion Fe fractions in coarse particles estimated by the model were often higher than the isotope-based estimates, especially near the Japanese coast (Table 1). A possible reason for this difference is the various characteristics of the "combustion Fe" that are included in the calculation. Combustion Fe estimated by the Fe isotope ratios is limited to Fe with low values of $\delta^{56}$Fe, which is emitted by evaporation under high-temperature conditions. Such evaporated Fe comprises aggregates of Fe (hydr)oxide particles mainly contained in fine-size particle fractions (Kurisu et al., 2016a, 2016b, 2019). In

contrast, the model estimates emissions from anthropogenic combustion sources based on the emission inventory dataset produced by the Community Emission Data System (Hoesly et al., 2018) and the Fe fraction for each particle size and from each emission source (Ito et al., 2018). Therefore, combustion Fe in the model-based estimation includes Fe emitted not only by evaporation processes but also by other processes, such as melting and fragmentation in coal combustion (Rathod et al., 2020), especially in coarse particles. This could have led to a larger contribution from combustion Fe to the coarse particles

near emission sources. Considering that the low $\delta^{56}$Fe corresponds to high fractional Fe solubility, it is suggested that evaporated Fe has high fractional Fe solubility and that Fe isotope ratios is important to estimate combustion Fe with high fractional solubility, although not all combustion Fe estimated by the model is included.

These results also indicate the importance of using the $\delta^{56}$Fe data of size-fractionated aerosols to evaluate the combustion Fe emission estimated by the model. To explain this, the $\delta^{56}$Fe values were reproduced according to Eq. (6) using

the $f_{combustion}$ estimated by the IMPACT model and were compared with the observational results (Fig. 10). When the $\delta^{56}$Fe values were reproduced for the bulk (coarse + fine) particles based on the results of the IMPACT model, they were much lower than the observed $\delta^{56}$Fe values, especially in group I samples. However, when calculating the values for fine particles only, they were better reproduced. Thus, the reduction in the combustion Fe emissions to fit to the observed higher bulk $\delta^{56}$Fe could mislead the model into underestimating the contribution of combustion Fe.

It should also be noted that the calculation of the fractions of combustion Fe largely depends on the $\delta^{56}$Fe of combustion Fe. Conway et al. (2019) conducted a similar comparison by applying −1.6 ‰ as the $\delta^{56}$Fe of combustion Fe, higher than that used in this study. They estimated 50 to 100% of the contribution of combustion Fe in soluble Fe in European and North American aerosols and suggested the underestimation of combustion Fe in a model calculation. As studies on the $\delta^{56}$Fe of combustion Fe are limited, further studies into the $\delta^{56}$Fe of combustion Fe from various sources and in different





regions will lead to a more quantitative understanding of the fraction of combustion Fe, which in turn will contribute to a more accurate prediction of the distribution of natural and combustion Fe in aerosols.

To further clarify the reasons for the difference between the observed and the model results, the Fe concentration of each source was compared with that estimated by the model. Although the bulk (fine + coarse) Fe concentrations were reproduced well by the model, the model underestimated the coarse Fe concentrations and overestimated the fine Fe

concentrations compared with those from observation (Fig. S10). This is partly because of the different size distribution of mineral dust from observation and the IMPACT model (Figs. S10a and b); the IMPACT model tended to underestimate the contribution of coarse mineral dust, as previously suggested by Adebiyi and Kok (2020) and Ito et al. (2020). The extent of overestimation and underestimation was larger when the Fe concentrations were low. Observational data of Fe concentrations and $\delta^{56}$Fe for size-fractionated aerosols in the open ocean will help improve the accuracy of the model estimation.


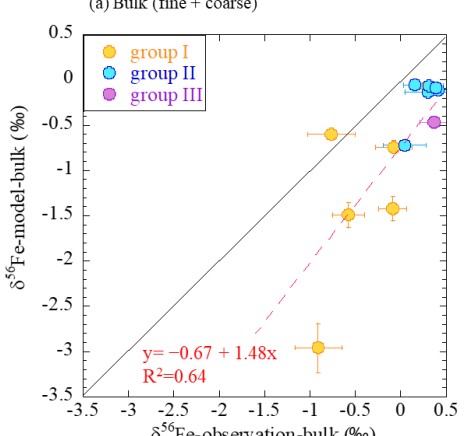

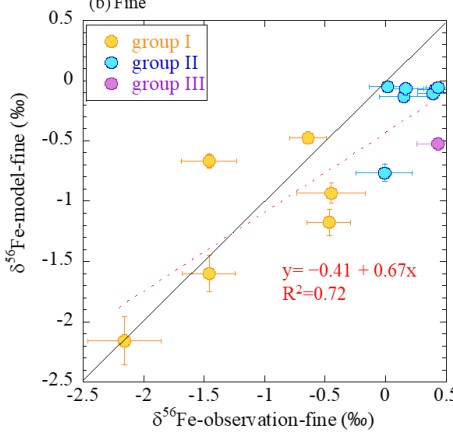

**Figure 10. Comparison of the $\delta^{56}$Fe values by observation with the IMPACT model prediction for (a) bulk (fine + coarse) and (b) fine particles. Yellow, group I; blue, group II; purple, group III.**





**4.5. Contribution of atmospheric Fe deposition to the surface seawater**

Our results suggest that Fe from anthropogenic combustion is an important source of soluble Fe in air masses from East Asia. The dry deposition fluxes ($F$) of total and soluble Fe from combustion and natural Fe were roughly estimated for each group according to the following equation:

$$F = C \times V, \tag{9}$$

where $C$ is the atmospheric Fe concentrations, in which the average total and soluble Fe concentrations in each group were used for the calculation. $V$ is the deposition rate of coarse and fine particles, which was assumed to be 1000 m day$^{-1}$ and 260 m day$^{-1}$, respectively, although these values can change depending on the wind speed and humidity (Buck et al., 2019; Duce and Tindale, 1991). To calculate the soluble Fe concentrations of the combustion and natural Fe in group I, the fractional Fe solubilities of combustion and natural Fe were estimated to be 11 % and 0.9 %, respectively, based on the values calculated from sample 13-e in section 4.3.

The deposition fluxes of total Fe in group I were approximately 201 nmol m$^{-2}$ day$^{-1}$ and 15 nmol m$^{-2}$ day$^{-1}$ for the natural and combustion sources, respectively (Table S7, Fig. 11), while those of soluble Fe were 2.9 nmol m$^{-2}$ day$^{-1}$ and 1.4 nmol m$^{-2}$ day$^{-1}$, respectively. Group I received a large contribution of combustion Fe (33 %) in the soluble Fe flux, suggesting the importance of combustion Fe as a source of soluble Fe in surface seawater despite its lower emission than natural sources. When considering the contribution of soluble Fe to the surface ocean, combustion and natural aerosol deposition could be important compared with other Fe sources, which are mainly transported from deeper layers of seawater, although this largely depends on the area and the season with a wide range of Fe flux from 0.5 to 20 nmol m$^{-2}$ day$^{-1}$ (Nishioka et al., 2007, 2020). Although there were no observational data for wet deposition in the present study, wet deposition flux has been shown to be larger than dry deposition flux in the western North Pacific (Uematsu et al., 1985). The IMPACT model also estimated the soluble Fe flux by dry and wet depositions during the group I sampling period, in which dry deposition accounted for up to 20 % of the total deposition flux (Fig. S11), indicating a much larger bulk (wet + dry) deposition flux of soluble Fe than the dry deposition only. If the contribution of combustion Fe to surface seawater is substantial, the low $\delta^{56}$Fe could be reflected in the surface seawater (Pinedo-gonzález et al., 2020). Simultaneous sampling and Fe isotope analysis of aerosols and surface seawater should allow the direct investigation of the importance of Fe in aerosols to the budged of soluble and insoluble Fe in the surface seawater.

The Fe deposition fluxes in groups II (27 and 0.2 nmol m$^{-2}$ day$^{-1}$ for total and soluble Fe, respectively) and III (32 and 0.4 nmol m$^{-2}$ day$^{-1}$ for total and soluble Fe, respectively) were only from natural sources. They were lower than those in group I by an order of magnitude (Table S7) and also considerably lower than those observed in previous studies (Buck et al., 2019; Duce and Tindale, 1991; Uematsu et al., 1983, 1985). Previous studies have reported the long-range transportation of a large quantity of mineral dust to the central North Pacific during spring (e.g., Buck et al., 2006; Uematsu et al., 1983). Therefore, if air masses are transported from East Asia, it is possible that the larger influence of atmospheric Fe, including combustion Fe, could occur more widely and also contributes to the increase in the supply of Fe to HNLC regions.





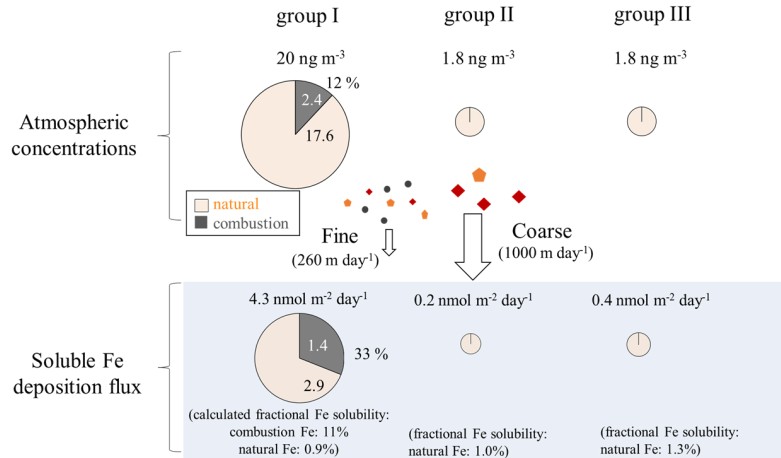

**Figure 11. Summary of the contribution of combustion and natural Fe in atmospheric aerosols and soluble Fe depositions.**

## 5. Conclusion

Size-fractionated aerosol samples were collected in the northwestern Pacific to estimate the relative contribution of anthropogenic combustion Fe and natural Fe to marine aerosols and to investigate the factors that control fractional Fe solubility. Air masses from East Asia included fine particles that yielded $\delta^{56}$Fe values 0.5 to 2 ‰ lower than those of the coarse particles because of the presence of combustion Fe. The $\delta^{56}$Fe values of coarse and fine particles in air masses from the eastern, central, or northern Pacific were close to the crustal value. It was also found that in air masses from East Asia, fractional Fe

solubilities are mainly controlled by the presence of combustion Fe. The proportion of combustion Fe in the total Fe in marine aerosols was up to 51 % and 20 % in fine and bulk (fine + coarse) particles, respectively. In addition, the contribution of combustion Fe was greater in the soluble component, suggesting the importance of combustion Fe as a source of Fe in the surface ocean. However, the influence of combustion Fe was limited in the vicinity of East Asia, and natural Fe was the main source of aerosol Fe in air masses from the eastern, central, or northern Pacific. The comparison of the contribution of

combustion Fe estimated by the IMPACT model with that estimated by the Fe isotope ratio suggested that $\delta^{56}$Fe values of the size-fractionated aerosol are important to make this comparison, due to the different characteristics of "combustion Fe" included in the different approaches, in which combustion Fe estimated using Fe isotope ratios is limited to particles emitted by evaporation mainly contained in fine particles. Considering that the $\delta^{56}$Fe values were correlated with fractional Fe solubilities, isotope-based estimation is important when discussing the contribution of combustion Fe with high fractional Fe

solubility. Although the influence of combustion Fe was limited adjacent to the Japanese coast in terms of samples in this study with relatively lower concentrations of Fe, the contribution may be much larger when air masses are transported to the open ocean from East Asia, such as during springtime in the northern hemisphere. This study showed the applicability of Fe isotope data to further understanding atmospheric Fe sources and their fractional Fe solubilities. The large contribution of combustion Fe to the marine atmosphere also suggests the possible contribution of combustion Fe to the surface seawater. Iron

isotope data will help in clarifying the contribution of combustion Fe to seawater; it is anticipated that this will lead to a more quantitative understanding of Fe cycling in the atmosphere-surface ocean system.



*Data availability*. The more detailed data are available upon request (Minako Kurisu, kurisum@jamstec.go.jp).

*Supplement.* The supplement related to this article is available online at XXXX.

*Author contributions.* MK and YT designed the research. MU established the sampling methods and collected aerosol samples during the cruises. MK, KS, and YT conducted sample analysis. AI developed the model code and performed the simulations. MK prepared the manuscript with contributions from all co-authors.


*Competing interests*. The authors declare that there is no conflict of interest.

*Acknowledgements.* We appreciate Tsuyoshi Iizuka for technical support with the isotope analysis. This study was supported by the Grant-in-Aid for JSPS Research Fellow Grant Number 17J06716 (to M.K.), JSPS KAKENHI Grant Number

18H04134 (to Y.T.) and 20H04329 (to A.I.), and Integrated Research Program for Advancing Climate Models (TOUGOU) Grant Number JPMXD0717935715 from the Ministry of Education, Culture, Sports, Science and Technology (MEXT), Japan (to A.I.). XAFS analysis was conducted with approval of KEK-PF (2016G632, 2018G575, 2018G089, 2019G093).

Figure captions.

**Figure 1**. Sampling areas in this study. The base figure was created using General Mapping Tools (GMT, Wessel et al., 2019)

**Figure 2**. Seven-day backward trajectories along the ship tracks for (a) group I, (b) group II, and (c) group III. The arrival height was set to 500 m and a new trajectory was obtained every 6 hours. The maps were created using GMT.

**Figure 3**. Atmospheric concentrations of (a) Fe and (b) soluble Fe, and (c) fractional solubility of Fe of fine and coarse particles during the KH-13-7 and KH-14-3 cruises. Errors were calculated from ICP-MS errors and blank subtraction errors. Yellow,

blue, and purple areas indicate the group I (air masses from the Asian continent), II (air masses from the central and eastern Pacific), and III (air masses from the northern North Pacific), respectively. <D.L., below the limit of detection limit due to higher concentrations in the blanks than samples; n.a., not available because the total Fe concentration was too low, * not analyzed due to insufficient sample amount was not enough.

**Figure 4**. Enrichment factors of (a) Fe, (b) V, and (c) Pb. Errors were calculated from ICP-MS error and blank subtraction

errors. Yellow, blue, and purple areas indicate group Is (air masses from the Asian continent), II (air masses from the central and eastern Pacific), and III (air masses from the northern North Pacific), respectively. *Not available due to insufficient sample.

**Figure 5**. Iron K-edge XANES spectra of (a) reference species, (b) KH-13-7 samples, (c) coarse particles of KH-14-3 samples, and (d) fine particles of KH-14-3 samples. (e) Fraction of each Fe species in the fine and coarse samples. Colored bars above

the graph (d) indicate the different air mass groups. *Not analyzed due to insufficient Fe. Yellow, blue, and purple areas indicate groups I (air masses from the Asian continent), II (air masses from the central and eastern Pacific), and III (air masses from the northern North Pacific), respectively.

**Figure 6**. Iron isotope ratios of KH-13-7 and KH-14-3 samples in (a) coarse and (b) fine particles. Data in the clear circles were not available. Values are also shown in Tables S5 and S6. The figures were produced using Ocean Data View (Schlitzer,

710    2020).

**Figure 7**. Scatter plots of Fe isotope ratios and (a) $EF_{Pb}$, (b) $EF_V$, and (c) CO. Concentrations of CO were averaged values during the sampling period (data from Kamezaki et al., 2019). $\rho$ is the Spearman Rank Order correlation coefficient ($p<0.05$). Yellow, group I; blue, group II; purple, group III.





**Figure 8**. The relationships between (a,b) the fractional Fe solubility and $\delta^{56}$Fe; (c, d) the fraction of Fe (hydr)oxides and
$\delta^{56}$Fe; and (e, f) the fraction of Fe (hydr)oxides and the fractional Fe solubility. $\rho$ is the Spearman Rank Order correlation
coefficient (p<0.05). *Only for fine particles. Yellow, group I; blue, group II; purple, group III.

**Figure 9**. The contribution of combustion Fe in (a) fine, (b) coarse, and (c) total (fine + coarse) Fe in marine aerosols. Data
with open circles were not determined.

**Figure 10**. Comparison of the $\delta^{56}$Fe values by observation with the IMPACT model prediction for (a) bulk (fine + coarse) and
(b) fine particles. Yellow, group I; blue, group II; purple, group III.

**Figure 11**. Summary of the contribution of combustion and natural Fe in atmospheric aerosols and soluble Fe depositions.

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
