# Peer review of "Contribution of combustion Fe in marine aerosols over the northwestern Pacific estimated by Fe stable isotope ratios"

_Atmospheric Chemistry and Physics, 2021_

## Author Response (AR1)

Revisions are highlighted in red color.

The revised figures and tables are shown following the replies to the comments.

**List of our replies to the comments kindly given by the Referee #1.**

| | | Comment | Reply and revision |
|---|---|---|---|
| 1 | | One of my concern is on the two different sets of the samples obtained by the two cruises: one obtained in winter (KH-13-7) and the other in summer (KH-14-3). The authors combined the data sets of the two cruises and classified into three groups by the trajectory analysis in terms of possible source regions. This classification is based on the emission area and transport pathways, while photochemical activities that may have affected photochemical and acidification processes (L. 458) during the transport of aerosols, are expected to be different between the periods of the two cruises. Is there any possibility that such difference in the photochemical fields of the atmosphere may have affected Fe fractional solubility, fraction of iron species, and Fe isotope ratios, even within the same group? I think the authors should add some more discussion on this point. | The average solar radiation fluxes were calculated along the backward trajectory using the HYSPLIT model, the results of which were added in Tables S1 and S2. They were substantially higher during the summer cruise than the winter one, but there was no systematic difference in fractional Fe solubility, fractions of Fe species, and Fe isotope ratios among the same group samples depending on the radiation flux (newly added as Fig. S9). We cannot deny the possibility that solar radiation is one of the factors affecting the Fe species and fractional Fe solubility, but in the present study, we found significant differences particularly for dataset classified by the different sources of air masses, instead of the photochemical fields of the atmosphere. However, we have added some discussion on the photochemical activities.

**Revision:**
L484. In addition, there was no relationship between the $\delta^{56}$Fe value and the average solar radiation flux calculated along the backward trajectory, suggesting the high $\delta^{56}$Fe values cannot be explained solely by the difference of photochemical activity (Fig. S9a), although we could not discuss the extent of the acidification process.

L514. A high solar radiation flux enhances photochemical reduction of Fe(III), especially under the presence of organic ligands, which can change fractional Fe solubilities and Fe species (Chen and Grassian, 2013; Pehkonen et al., 1993). Because KH-13-7 and KH-14-3 samples were collected during winter and summer, respectively, their solar radiation fluxes were substantially different (Tables S1 and S2). However, we did not find any effects by the solar radiation fluxes on fractional Fe solubilities and Fe species between the winter (KH-13-7) and the summer (KH-14-3) samples (Figs. S9b and S9c). |

| 2 | I suppose that the fractions of group I, II, and III in each cruise samples reflect the difference in the metrological conditions (wind fields) as well as the differences in the geographical locations where the aerosol samplings were made between winter (KH-13-7) and summer (KH-14-3). If this is true, I think the authors should mention it in the text. | In group I, air masses were transported far from the western part of the Eurasian continent over East Asia in winter (KH-13-7), whereas they were transported over Japan or the western part of the Pacific in summer (KH-14-3). This difference could lead to the different Fe concentrations and Fe isotope ratios in KH-13-7 and KH-14-3 samples. In group II, the air masses were consistently from the east for both winter and summer samples. All of the group III samples were collected in the summer. We added some discussion on the different Fe concentrations and Fe isotope ratios among the group I samples in the text. In addition, we revised Fig. 2 so that the difference between the two cruises could be seen. |
| | | **Revision:** |
| | | L. 262: The first group was mainly transported from the direction of East Asia via westerly winds and corresponded to samples 13-a, 13-d, 13-e, 14-A, 14-N, and 14-O (group I, Fig. 2a). Winter samples (KH-13-7) were transported longer distances from the Eurasian continent, whereas summer samples (KH-14-3) passed over Japan or the western part of the Pacific. They possibly contained aerosols emitted both by natural and anthropogenic sources. |
| | | L. 270: Among the group I samples, KH-13-7 samples showed high concentrations even when the sampling points were relatively far from the Japanese coast. This is possibly because they contained a large number of particles transported from the Eurasian continent. |
| | | L. 555: The fractions in the group I samples were higher in the summer (KH-14-3, on average 20 %) than in the winter (KH-13-7, on average 6 %), which suggests that the influence of mineral dust from East Asia was greater during the winter (KH-13-7) than the summer (KH-14-3). |
| 3 | L.328: "Fe oxides were only found in the group III samples," while they are dominant iron species identified in the group III (Figure 5e). Is this attributable to the oceanic region where the aerosols were sampled, or is this | Possibilities of the dominance of Fe oxides only in group III are different sources or different processes during transportation. Iron oxides are often found in soils and aerosols on land (Schroth et al., 2009; Kurisu |

related to the specific season? Not only saying "different sources," but also additional discussion on possible sources or processes of this fraction should be made.

et al., 2016; Ito and Wagai, 2017; Lu et al., 2017). Although air masses were transported via the eastern part of Russia, the Kamchatka Peninsula, and Alaska, the soils in these regions are not enriched with Fe oxides compared with those in the Eurasian continent, such as China and Japan (Ito and Wagai, 2017; Lu et al., 2017). Therefore, different dust sources are possibly not the reason for the dominance of Fe oxides only in group III. Ship emission could be another source of Fe considering that some of the coarse particles in group III had $EF_V$ higher than 10. However, oil fly ash often contains Fe(III) sulfate and Fe oxides are not dominant (Schroth et al., 2009). According to the volcanic eruption database (Global Volcanism Program, Smithsonian Institution, https://volcano.si.edu/), there were several volcanic activity events around the Aleutian Islands during the sampling periods. Although aerosols emitted by volcanic activities also contain Fe oxides (Ayris and Delmelle, 2012), they are not dominant as observed in group III samples.

If the different process during the transport is the reason, it is possible that crystalline Fe oxides in group III remained without reacting heavily with acids, since crystalline Fe oxides are more slowly dissolved with acids compared with other Fe species, such as ferrihydrite and Fe-containing silicates (Journet et al., 2009).

It is also possible that the results are related to the specific season (summer) and regions. However, we did not collect aerosols in winter, and thus it is difficult to draw any firm conclusions from it.

Based on the discussion above, we cannot pin down the clear reason why Fe oxides were dominant only in the group III samples.

In the revised paper, we combined ferrihydrite and Fe oxides fractions, which we named as Fe (hydr)oxides fractions as in Fig. 5. Because we have already combined them in the following section (4.2), the combination of the two fractions in Fig. 5 makes our discussion smoother.

**Revision:**
L. 353: For simplification, the main Fe species were categorized into Fe-containing silicates (fayalite, weathered biotite, chlorite, and illite with various Fe(II) fractions) and Fe (hydr)oxides (magnetite, goethite, hematite, and ferrihydrite, Fig. 5e),

| | | | |
|---|---|---|---|
| | | | considering that Fe-containing silicates are of natural origin, whereas Fe (hydr)oxides can be both of natural and combustion origin. |
| 4 | In the text, it would be helpful to show the averages and standard deviations of the concentrations, fractional solubility, and isotope ratios of Fe in each group (I, II, and III) in addition to the ranges. I understand that the number of data would become limited in each classification, but still the average values and the variation provide some basic information. | | Thank you for your suggestion. We added the averages and standard deviations of the concentrations, fractional Fe solubility, EF, and Fe isotope ratios in each group. Based on the comment from Reviewer 2, we also added some discussion on EF.

**Revision:**
L. 272: The Fe concentration was significantly affected by the source of air masses. The bulk Fe concentrations (fine+coarse) were 2.6 to 41.8 ng m$^{-3}$ in the vicinity of East Asia (group I, 19.8±13.8 (SD) ng m$^{-3}$).

L. 272: Samples in groups II (1.97±1.48 ng m$^{-3}$) and III (1.78±0.68 ng m$^{-3}$) showed Fe concentrations of less than 5 ng m$^{-3}$ (Fig. 3a, Tables S4 and S5).

L. 288: The average values (±SD) were 1.52±0.48 (group I of the coarse particles), 1.27±0.56 (group II of the coarse particles), 0.98±0.08 (group III of the coarse particles), 1.23±0.35 (group I of the fine particles), and 1.20±0.73 (group II of the fine particles), respectively. Their values were close to 1, suggesting that Fe mainly originated from crustal sources. When we compared the difference among the groups, a significant difference between group I and III for the coarse particles was found and was not found in the other groups (by the Wilcoxon rank-sum test, significant difference at $p<0.05$). The higher $EF_{Fe}$ value of the coarse particles of group I than group III indicates that group I contained Fe-rich particles. Considering that the high $EF_{Fe}$ values were observed near the Japanese coast (14-A and 14-O), coarse particles of group I possibly contained some amount of anthropogenic Fe emitted by industrial activities in Japan, in addition to crustal materials. The influence of combustion Fe in the other groups was not clear only from the $EF_{Fe}$.

L. 297: The $EF_{Pb}$ of the fine particles in group I ($EF_{Pb}$=764±739 (SD)) was higher than that of the other samples (203±142 and 161±23 for groups II and III, respectively), especially in the winter (KH-13-7, $EF_{Pb}$=1289±727), which is indicative of a large |

influence from coal combustion in East Asia. $EF_V$ values varied from 1.7 to 840, with higher $EF_V$ values in fine particles ($EF_V$=74±185) than in coarse particles (Fig. 4b, $EF_V$=4.92±3.82). The $EF_V$ values of the fine particles in the group III samples ($EF_V$=370±320) were more than 100, higher than the other samples, which was not the case for Pb.

L. 328: The average fractional Fe solubility (±SD) of each group was 1.46±1.51 %, 1.13±1.62 %, and 2.22±1.43 % for the coarse particles of the groups I, II, and III, respectively, whereas 5.89±4.04 % and 4.52±6.69 % for the fine particles of the groups I and II, respectively. Although there were no clear differences in the fractional solubilities of the different air mass groups (by the Wilcoxon rank-sum test, significant difference at p<0.05), fine particles often showed higher soluble Fe concentration and fractional Fe solubility than the coarse particles of the same sample, which is consistent with previous reports (Kurisu et al., 2016b; Ooki et al., 2009).

L. 375: Negative $\delta^{56}Fe$ values ranging from −0.45 to −2.16 ‰ were observed in the fine particles of group I (on average −1.10±0.63 ‰ (SD)), in air masses from the direction of East Asia (Fig. 6, Tables S4 and S5). These values were much lower than those observed for the coarse particles (on average −0.02±0.12 ‰). The $\delta^{56}Fe$ values were particularly low in the fine particles of samples collected in the vicinity of the Japanese coastline (13-a, 14-N, and 14-O). The bulk (coarse + fine) $\delta^{56}Fe$ values calculated from the Fe concentration and $\delta^{56}Fe$ of each size fraction ranged from −0.07 to −0.91 ‰ (Tables S5 and S6, on average −0.45±0.32 ‰), which is considerably lower than the values obtained from North American or European air masses in the North Atlantic, which reached as low as −0.16 ‰ (Conway et al., 2019), suggesting an important contribution of aerosols with low $\delta^{56}Fe$ values in the North Pacific aerosols. These low $\delta^{56}Fe$ values may have originated from combustion Fe, which is discussed in more detail later.

    In air masses from the central, eastern, or northern Pacific (in groups II and III), the $\delta^{56}Fe$ values of both the coarse (on average 0.25±0.14 ‰ and 0.14±0.10 ‰ for group II and III, respectively) and fine (on average 0.23±0.17 ‰ for group II samples

| | | | |
|---|---|---|---|
| | | | and 0.43±0.17 ‰ for 14-L in group III) particles were close to or higher than 0 ‰. The coarse and fine particles in each sample yielded similar $\delta^{56}Fe$ values to each other, although the $\delta^{56}Fe$ of the fine particles in some samples could not be measured due to the low quantity of sample Fe compared with blank Fe. |
| 5 | The authors show the term "East Asia" quite often (e.g., six times in the abstract). Maybe they intend to say that East Asia is characterized by anthropogenic sources, while they also discuss possible effect of biomass burning in Siberia which is not included in East Asia. According to Figure 2(a), some air masses are coming from far west and passed over eastern part of the Eurasian continent, while some pathways in the south of 30 degree north are not clear to me. Please briefly characterize "East Asia" in terms of sources (terrestrial natural source is also expected) and clarify the pathways of the trajectory in Figure 2(a). | | We intended to explain that group I samples were transported mainly from "the direction of" East Asia, in which both anthropogenic and terrestrial natural sources are contained. Although some of the trajectories did not pass over the Eurasian continent but the ocean, we categorized the sample into group I because most of the trajectories passed over land. As you pointed, air masses of the group I were also passed over the eastern or western part of the Eurasian continent. We changed the term "East Asia" to "the direction of East Asia" and added some explanation on the air mass sources of group I.

**Revision:**
L.262: The first group was mainly transported from the direction of East Asia via westerly winds and corresponded to samples 13-a, 13-d, 13-e, 14-A, 14-N, and 14-O (group I, Fig. 2a). Winter samples (KH-13-7) were transported longer distances from the the Eurasian continent, whereas summer samples (KH-14-3) passed over Japan or the western part of the Pacific. They possibly contained aerosols emitted both by natural and anthropogenic sources. |
| 6 | L.360: "..suggesting an importance of aerosols with low $\delta^{56}Fe$ values": importance in terms of what? | | We intended to suggest that there was a larger contribution of aerosols with low $\delta^{56}Fe$ (i.e. combustion Fe) in the North Pacific aerosols than in the North Atlantic ones.

**Revision:**
L. 381: …suggesting an important contribution of aerosols with low $\delta^{56}Fe$ values in the North Pacific aerosols. |
| 7 | The word "transportation" generally refers to a system or method for carrying passengers or goods by a vehicle or a vessel or an airplane. For atmospheric aerosol or air masses, "transport" is generally used and is more appropriate word. | | Thank you for pointing out the error in my English writing.
We changed the word "transportation" to "transport." |
| 8 | Figure 11: The authors should mention what the size of the pie chart indicates. | | The size of the pie chart indicates the concentrations and fluxes. We added the |

| | |
|---|---|
| Maybe the magnitude of the concentrations and flux? | explanation in the figure caption.

**Revision:**
Figure 11. Summary of the contribution of combustion and natural Fe in atmospheric aerosols and soluble Fe depositions. The sizes of the pie charts are indicative of the magnitude of the atmospheric Fe concentrations and soluble Fe deposition fluxes. |

**List of our replies to the comments kindly given by the Referee #2.**

| | Comment | Reply and revision |
|---|---|---|
| 1 | Line 91: In reference to Myriokefalitakis et al (2018), the authors state, "such as the relative fraction of combustion and dust Fe to the soluble Fe that is present over oceanic regions...". I believe this intends to refer to the relative contribution of combustion and dust Fe to the soluble Fe. The term fraction was a bit confusing in this context. | Thank you for your suggestion. We changed the term "fraction" to "contribution."

 **Revision:**
 L. 92: They claimed that some uncertainties remain, such as the relative contribution of combustion and dust Fe to the soluble Fe that is present over remote oceanic regions…. |
| 2 | Line 104-105: It is not clear what is meant by "which is not directly associated with the observed T-Fe and S-Fe concentrations". I believe the authors mean that we are unable to determine the relative contributions of combustion Fe and natural Fe (assumed to mean mineral dust). Suggest rewording to "which is not possible from the observed T-Fe and S-Fe concentrations". | As you suggested, we wanted to explain the difficulty of determining the relative contribution of Fe from different sources.

 **Revision:**
 L. 105: .., which is not possible from the observed T-Fe and S-Fe concentrations,… |
| 3 | Line 142: Were certified reference materials digested and analyzed to assess the efficacy of the digests? This data should be included if available. | We measured the certified reference material (simulated Asian mineral dust, CJ-2). We added the measured values in the supporting information (Table S3).

 **Revision:**
 L. 159: The effectivity of the measurement was confirmed by measuring the certified reference material CJ-2 (Simulated Asian mineral dust, Table S3; Nishikawa et al., 2000) |
| 4 | Line 150: >18.2 MΩ·cm | Thank you for your correction.

 **Revision:**
 L. 151: (> 18.2 MΩ cm, Milli-Q, Millipore GmbH, Japan) |
| 5 | Line 156: What Fe isotope was measured by the quadrupole ICPMS? Were any measures taken to remove polyatomic interferences? | We measured $^{56}$Fe in helium collision cell mode to remove ArO interference.

 **Revision:**
 L. 158: For the measurement of $^{56}$Fe, helium collision cell mode was used to remove ArO interference. |
| 6 | Line 274: What were the sources of the large errors? | The amount of Pb and Ti on the filter was small, especially in KH-14-3 samples. We could |

| | | |
|---|---|---|
| | | use the limited amount of the sample for the ICP-MS measurement and their concentrations were close to their detection limit, which made the errors large. In addition, a relatively high concentration of Ti and Pb on a blank filter compared to the sample themselves also affected the error. |
| 7 | Line 279: How were the EF values tested to determine if there statistical differences across the groups? | We had only qualitatively compared them, so we compared their average values and also conducted the Wilcoxon rank-sum test. We did not use the t-test because the data did not follow a normal distribution. As a result, there was a difference between the EF of the coarse particles of the group I and III, whereas there was no significant difference among the other groups. We added some discussions on the difference.

**Revision:**
L. 288: The average values (±SD) were 1.52±0.48 (group I of the coarse particles), 1.27±0.56 (group II of the coarse particles), 0.98±0.08 (group III of the coarse particles), 1.23±0.35 (group I of the fine particles), and 1.20±0.73 (group II of the fine particles), respectively. Their values were close to 1, suggesting that Fe mainly originated from crustal sources. When we compared the difference among the groups, a significant difference between group I and III for the coarse particles was found and was not found in the other groups (by the Wilcoxon rank-sum test, significant difference at $p<0.05$). The higher $EF_{Fe}$ value of the coarse particles of group I than group III indicates that group I contained Fe-rich particles. Considering that the high $EF_{Fe}$ values were observed near the Japanese coast (14-A and 14-O), coarse particles of group I possibly contained some amount of anthropogenic Fe emitted by industrial activities in Japan, in addition to crustal materials. The influence of combustion Fe in the other groups was not clear only from the $EF_{Fe}$.

L. 607: Actually, coarse particles with relatively high $EF_{Fe}$ values and $\delta^{56}Fe$ close to 0 ‰ observed near the Japanese coast suggest the presence of anthropogenic Fe which was not emitted by evaporation. |
| 8 | Line 299 Fig3 caption: Delete "was not enough" | We deleted the words. |
| 9 | Line 380: change "sorely" to "solely" | Thank you for pointing out the error. We corrected the word. |
| 10 | Line 398-399: The final sentence does not fit here. The same statement follows | We agree with your suggestion. We deleted the sentence. |

| | | |
|---|---|---|
| | at Line 421 after it has been more fully justified. Suggest delete the first use where the claim appear overly confident. | |
| 11 | Line 459: Parantheses around acidic are not necessary | We deleted the parentheses. |
| 12 | Line 560: Perhaps remind the reader what Comp 3 refers to. | We added the explanation for Comp 3.

**Revision:**
L. 587: The high proportions of combustion Fe in the fine particles in group I were reproduced mainly by Comp 3 (anthropogenic combustion on land, mainly from coal combustion and steel industry) in the IMPACT model, … |
| 13 | Line 581-582: Change both uses of "is" to "are" for correct subject/verb agreement. | We changed "Fe isotope ratios" to "the $\delta^{56}$Fe." We also added "the contribution of" to make it easier to understand the sentence.

**Revision:**
L. 609: …, it is suggested that evaporated Fe has high fractional Fe solubility and that the $\delta^{56}$Fe is important to estimate the contribution of combustion Fe with high fractional solubility, although not all combustion Fe estimated by the model is included. |

**Revised Figures and Tables (Figs. 2, 5; Tables S1-S3, S7; Fig. S9)**

[Figure]

**Figure 2.** Seven-day backward trajectories along the ship tracks for (a) group I, (b) group II, and (c) group III. The arrival height was set to 500 m and a new trajectory was obtained every 12 hours. The maps were created using GMT. Red and blue lines indicate trajectories during the KH-13-7 and 14-3 cruises, respectively.

[Figure]

**Figure 5**. Iron K-edge XANES spectra of (a) reference species, (b) KH-13-7 samples, (c) coarse particles of KH-14-3 samples, and (d) fine particles of KH-14-3 samples. (e) Fraction of each Fe species in the fine and coarse samples. Colored bars above the graph (d) indicate the different air mass groups. *Not analyzed due to insufficient Fe. Yellow, blue, and purple areas indicate groups I (air masses from the direction of East Asia), II (air masses from the central and eastern Pacific), and III (air masses from the northern North Pacific), respectively.

**Table S1:** Aerosol sampling periods of KH-13-7 cruise. Solar radiation flux is the average flux along the backward trajectory calculated by the HYSPLIT model (Stein et al., 2015).

| No. | Start time (UTC) | Start point | Total Flow ($m^3$) | Solar radiation flux ($W\ m^{-2}$) |
|---|---|---|---|---|
| | End time (UTC) | End point | | |
| 13-a | 12 Dec, 11:04 p.m. | 29.22º N, 147.92º E | 923.14 | 149.7 |
| | 15 Dec, 10:00 p.m. | 20.00º N, 160.00º E | | |
| 13-b | 15 Dec, 10:05 p.m. | 20.00º N, 160.00º E | 717.09 | 243.3 |
| | 18 Dec, 9:01 p.m. | 11.80º N, 172.28º E | | |
| 13-c | 2 Feb, 10:12 p.m. | 3.67º N, 159.44º E | 734.02 | 256.3 |
| | 5 Feb, 11:05 p.m. | 19.39º N, 150.31º E | | |
| 13-d | 5 Feb, 11:08 p.m. | 19.39º N, 150.31º E | 605.99 | 158.5 |
| | 9 Feb, 0:03 a.m. | 29.62º N, 142.33º E | | |
| 13-e | 9 Feb, 0:05 a.m. | 29.62º N, 142.33º E | 597.42 | 122.7 |
| | 11 Feb, 0:03 a.m. | 35.17º N, 139.77º E | | |

**Table S2.** Aerosol sampling periods of KH-14-3 cruise. Solar radiation flux is the average flux along the backward trajectory calculated by the HYSPLIT model (Stein et al., 2015).

| No. | Start time (UTC) | Start point | Total Flow (m$^3$) | Solar radiation |
| --- | --- | --- | --- | --- |
| | End time (UTC) | End point | | flux (W m$^{-2}$) |
| 14-A | 24 June, 0:13 a.m. | 32.40° N, 143.09° E | 710.64 | 386.9 |
| | 26 June, 10:04 p.m. | 22.00° N, 157.26° E | | |
| 14-B | 26 June, 10:06 a.m. | 22.00° N, 157.26° E | 468.72 | 376.3 |
| | 28 June, 1:08 p.m. | 17.56° N, 163.05° E | | |
| 14-C | 30 June, 10:04 p.m. | 10.33° N, 174.18° E | 887.42 | 345.7 |
| | 3 July, 8:00 p.m. | 1.24° N, 171.45° W | | |
| 14-D | 3 July, 8:04 p.m. | 1.24° N, 171.45° W | 818.32 | 331.4 |
| | 6 July, 7:55 p.m. | 10.02° N, 170.02° W | | |
| 14-E | 6 July, 7:57 p.m. | 10.02° N, 170.02° W | 864.71 | 381.7 |
| | 9 July, 8:20 p.m. | 20.01° N, 169.57° W | | |
| 14-F | 9 July, 8:22 p.m. | 20.01° N, 169.57° W | 871.97 | 390.3 |
| | 12 July, 6:59 p.m. | 20.46° N, 161.33° W | | |
| 14-G | 12 July, 7:02 p.m. | 20.46° N, 161.33° W | 228.41 | 390.5 |
| | 13 July, 3:57 p.m. | 21.01° N, 157.59° W | | |
| 14-H | 18 July, 4:13 a.m. | 21.51° N, 157.36° W | 345.16 | 392.2 |
| | 20 July, 8:01 p.m. | 30.02° N, 170.01° W | | |
| 14-I | 20 July, 8:03 p.m. | 30.02° N, 170.01° W | 898.01 | 391.3 |
| | 23 July, 7:58 p.m. | 40.03° N, 171.00° W | | |
| 14-J | 23 July, 8:00 p.m. | 40.03° N, 171.00° W | 933.27 | 383.5 |
| | 26 July, 8:00 p.m. | 50.02° N, 169.56° W | | |
| 14-K | 26 July, 8:02 p.m. | 50.02° N, 169.56° W | 944.25 | 371.2 |
| | 29 July, 8:57 p.m. | 63.38° N, 167.38° W | | |
| 14-L | 29 July, 9:01 p.m. | 63.38° N, 167.38° W | 748.4 | 364.4 |
| | 1 Aug, 8:59 p.m. | 60.44° N, 176.03° W | | |
| 14-M | 1 Aug, 9:00 p.m. | 60.44° N, 176.03° W | 954.26 | 371.4 |
| | 4 Aug, 9:57 p.m. | 47.39° N, 167.08° E | | |
| 14-N | 4 Aug, 10:00 p.m. | 47.39° N, 167.08° E | 926.04 | 377.0 |
| | 8 Aug, 0:05 a.m. | 37.33° N, 145.15° E | | |
| 14-O | 8 Aug, 0:07 a.m. | 37.33° N, 145.15° E | 293.43 | 375.8 |
| | 8 Aug, 11:59 p.m. | 35.26° N, 139.48° E | | |

**Table S3**. Comparison of elemental concentrations (µg/g) of CJ-2 (Simulated Asian mineral dust) measured in this study (±SD, n=3) and Nishikawa et al. (2000).

|  | Ti | V | Fe | Pb |
|---|---|---|---|---|
| measured value (±SD) | 4400±110 | 67.4±18.4 | 29600±2500 | 21.3±6.2 |
| Nishikawa et al. (2000) | 4600 | 77 | 30200 | 21.2 |

**Table S7.** The fraction (%) of each Fe species estimated by the linear combination fitting of XANES spectra. n.a.: not available due to small quantity of sample Fe or high filter blank.

| No. | Coarse | | | Fine | | |
|---|---|---|---|---|---|---|
| | Fe-containing aluminosilicates | Fe (hydr)oxides | $R$ | Fe-containing aluminosilicates | Fe (hydr)oxides | $R$ |
| 13-a | 59 | 41 | 0.013 | 34 | 66 | 0.012 |
| 13-b | 55 | 45 | 0.026 | 43 | 57 | 0.028 |
| 13-c | 21 | 79 | 0.013 | 20 | 80 | 0.029 |
| 13-d | 42 | 58 | 0.012 | 32 | 68 | 0.012 |
| 13-e | 66 | 34 | 0.013 | 52 | 48 | 0.017 |
| 14-A | 37 | 63 | 0.014 | 42 | 58 | 0.021 |
| 14-B | 77 | 23 | 0.022 | 33 | 67 | 0.029 |
| 14-C | 40 | 60 | 0.016 | 36 | 64 | 0.044 |
| 14-D | | n.a. | | | n.a. | |
| 14-E | 35 | 65 | 0.011 | 27 | 73 | 0.028 |
| 14-F | 36 | 64 | 0.015 | 33 | 67 | 0.029 |
| 14-G | 52 | 48 | 0.040 | 44 | 56 | 0.028 |
| 14-H | 56 | 44 | 0.008 | 24 | 76 | 0.017 |
| 14-I | 82 | 18 | 0.011 | | n.a. | |
| 14-J | 21 | 79 | 0.013 | | n.a. | |
| 14-K | 23 | 77 | 0.015 | | n.a. | |
| 14-L | 27 | 73 | 0.014 | | n.a. | |
| 14-M | 36 | 64 | 0.022 | | n.a. | |
| 14-N | 54 | 46 | 0.020 | 36 | 64 | 0.019 |
| 14-O | 35 | 65 | 0.018 | 12 | 88 | 0.054 |

[Figure]

**Figure S9.** The relationship between the average solar radiation flux and (a) $\delta^{56}$Fe, (b) fractional Fe solubility, and (c) fraction of Fe (hydr)oxides. Yellow: group I; blue: group II; purple: group III.